# Hyperpolarised 13C-MRI identifies the emergence of a glycolytic cell population within intermediate-risk human prostate cancer

Nikita Sushentsev [1], Mary A. McLean[1,2], Anne Y. Warren[3], Arnold J. V. Benjamin [1], Cara Brodie[2], Amy Frary [1], Andrew B. Gill[1], Julia Jones [2], Joshua D. Kaggie [1], Benjamin W. Lamb [4,5], Matthew J. Locke[1], Jodi L. Miller [2], Ian G. Mills[6,7,8,9], Andrew N. Priest [1], Fraser J. L. Robb[10], Nimish Shah[4], Rolf F. Schulte[11], Martin J. Graves [1], Vincent J. Gnanapragasam[4,12,13], Kevin M. Brindle [2,14], Tristan Barrett[1,15 ✉] & Ferdia A. Gallagher [1,15]

Hyperpolarised magnetic resonance imaging (HP 13C-MRI) is an emerging clinical technique to detect [1-13C]lactate production in prostate cancer (PCa) following intravenous injection of hyperpolarised [1-13C]pyruvate. Here we differentiate clinically significant PCa from indolent disease in a low/intermediate-risk population by correlating [1-13C]lactate labelling on MRI with the percentage of Gleason pattern 4 (%GP4) disease. Using immunohistochemistry and spatial transcriptomics, we show that HP 13C-MRI predominantly measures metabolism in the epithelial compartment of the tumour, rather than the stroma. MRI-derived tumour [1-13C]lactate labelling correlated with epithelial mRNA expression of the enzyme lactate dehydrogenase (LDHA and LDHB combined), and the ratio of lactate transporter expression between the epithelial and stromal compartments (epithelium-to-stroma MCT4). We observe similar changes in MCT4, LDHA, and LDHB between tumours with primary Gleason patterns 3 and 4 in an independent TCGA cohort. Therefore, HP 13C-MRI can metabolically phenotype clinically significant disease based on underlying metabolic differences in the epithelial and stromal tumour compartments.

[1] Department of Radiology, Addenbrooke's Hospital and University of Cambridge, Cambridge, UK. [2] Cancer Research UK Cambridge Institute, University of Cambridge, Cambridge, UK. [3] Department of Pathology, Cambridge University Hospitals NHS Foundation Trust, Cambridge, UK. [4] Department of Urology, Cambridge University Hospitals NHS Foundation Trust, Cambridge, UK. [5] School of Allied Health, Anglia Ruskin University, Cambridge, UK. [6] Patrick G Johnston Centre for Cancer Research, Queen's University Belfast, Belfast, UK. [7] Nuffield Department of Surgical Sciences, University of Oxford, John Radcliffe Hospital, Oxford, UK. [8] Centre for Cancer Biomarkers, University of Bergen, Bergen, Norway. [9] Department of Clinical Science, University of Bergen, Bergen, Norway. [10] GE Healthcare, Aurora, OH, USA. [11] GE Healthcare, Munich, Germany. [12] Division of Urology, Department of Surgery, University of Cambridge, Cambridge, UK. [13] Cambridge Urology Translational Research and Clinical Trials Office, Cambridge Biomedical Campus, Addenbrooke's Hospital, Cambridge, UK. [14] Department of Biochemistry, University of Cambridge, Cambridge, UK. [15] These authors contributed equally: Tristan Barrett, Ferdia A. Gallagher. ✉email: tb507@medschl.cam.ac.uk

Prostate cancer (PCa) is the second commonest and the fifth deadliest male cancer worldwide with the global burden of disease expected to double by 2040[1]. The use of serum prostate-specific antigen (PSA) testing has led to overdiagnosis of non-aggressive disease, with 48 cases of PCa needing treatment to avert one death[2,3]. Conversely, clinically significant PCa may be undetected using screening modalities such as PSA and systematic biopsy, with more than half of patients continuing to present with locally advanced and/or metastatic PCa[4–6]. Furthermore, 27% of PCa patients enroled on active surveillance with assumed indolent disease show histopathological disease progression during the first 5 years[7]. Consequently, there is a pressing need to develop accurate and accessible risk-stratification tools to differentiate indolent from clinically aggressive disease.

The recent international adoption of pre-biopsy magnetic resonance imaging (MRI) as the first-line investigation in patients with suspected clinically localised PCa[8] has created a paradigm shift in the PCa diagnostic pathway[9]. MRI has a negative predictive value above 90%[10], avoids unnecessary biopsies in ~50%[11], and reduces the over-detection of indolent disease by ~13%[12]. However, MRI has a relatively poor positive predictive rate for detecting clinically significant PCa of ~35%[13], meaning histopathological confirmation is typically required, which in turn is limited by the potential for sampling error and procedure-associated complications[14]. Therefore, improving the diagnostic performance of MRI and its specificity for aggressive disease could reduce the requirement for tissue sampling and associated patient morbidity.

Imaging the metabolic alterations that occur during tumour development is a promising approach for improving the diagnostic potential of MRI. The Warburg effect is characterised by lactate production even in normoxic conditions, which in turn generates an acidic extracellular microenvironment that stimulates metastasis[15], promotes several metabolic pathways that facilitate cellular proliferation, and provides a fuel source for some cell subpopulations within the tumour[16,17]. Hyperpolarised [1-¹³C]pyruvate MRI (HP ¹³C-MRI) is an emerging clinical imaging technique[18] that can probe the exchange of the hyperpolarised ¹³C label between pyruvate and lactate, catalysed by the enzyme lactate dehydrogenase (LDH)[19]. The first-in-human study of the technique demonstrated elevated [1-¹³C]lactate production in localised PCa[20], with more recent studies demonstrating a relationship between Gleason grade and increasing lactate labelling[21], as well as demonstrating a potential role for imaging metastatic PCa[22]. Therefore, HP ¹³C-MRI is a promising tool for addressing the diagnostic challenges presented by PCa.

Normal epithelial cells in the prostate are uniquely characterised by high levels of aerobic glycolysis and low levels of oxidative metabolism. This unusual metabolic phenotype is driven by the requirement to export citrate into the seminal fluid that comes at the expense of truncating flux through the tricarboxylic (TCA) cycle as a result of zinc-induced inhibition of m-aconitase[23]. Conversely, early-stage PCa cells exhibit a marked decrease in zinc concentration, which has been linked to the restoration of TCA cycle activity and oxidative metabolism[24]. A number of preclinical and clinical studies have shown that early-stage PCa cells may fuel TCA cycle activity by importing lactate from neighbouring cancer-associated fibroblasts (CAFs)[25–28], a phenomenon also observed in other tumour types[29,30]. This finding is underpinned by cell-specific expression of the monocarboxylate transporters 1 and 4 (MCT1/4), where MCT1 and MCT4 are expressed mostly on tumour and stromal cells respectively and mediate lactate/pyruvate influx (MCT1) and lactate efflux (MCT4)[28,31,32]. In contrast, aggressive prostate tumours and metastatic lesions demonstrate higher glycolytic flux, driven partly by hypoxia-induced upregulation of glycolytic

enzymes[33], and partly by the metabolic reprogramming that results from genomic abnormalities, such as loss of PTEN[34,35]. Therefore, higher glycolytic flux in high-grade tumours[36] might explain the increased [1-¹³C]lactate labelling in aggressive disease compared to more oxidative early-stage lesions. HP ¹³C-MRI offers the potential to non-invasively probe these changes in tumour metabolism over time, and their early detection could be used to differentiate indolent from aggressive disease.

This study investigated the role of HP ¹³C-MRI in patients with low, intermediate, and high-risk PCa, to assess differences in tumour metabolic phenotype. We compared this novel metabolic imaging to standard-of-care measures of tumour aggressiveness, immunohistochemical (IHC) expression of the transporters for pyruvate and lactate (MCT1 and MCT4), and mRNA expression of the enzyme subunits catalysing the exchange between the two metabolites (LDHA and LDHB). We demonstrate correlations between hyperpolarised [1-¹³C]lactate labelling, percentage of Gleason pattern 4 (%GP4) disease, proton (¹H) MRI-derived apparent diffusion coefficient (ADC) as a measure of cellularity, the number of tumour epithelial cells measured on histology, combined epithelial LDH expression, and the epithelium-to-stroma MCT4 ratio. We also tested these findings using an independent dataset from The Cancer Genome Atlas (TCGA) to provide a potential biological explanation for the increase in HP [1-¹³C]lactate labelling in more aggressive human PCa, which reflects the increasing glycolytic flux in tumour epithelial cells.

## Results

Ten patients with biopsy-proven PCa underwent HP ¹³C-MRI prior to robot-assisted radical prostatectomy (RARP) (Supplementary Fig. 1). Post-surgical histopathological assessment of whole-mount slides revealed the presence of 15 lesions (Supplementary Table 1). Two tumours were excluded from the imaging analysis due to technical failure of the HP ¹³C-MRI (one illustrated in Supplementary Fig. 2). Metabolic parameters were derived from 13 tumours (Table 1), nine of which showed International Society of Urological Pathology (ISUP) grade group 2 disease. Tumours were subdivided based on %GP4 as a more refined pathological biomarker of tumour aggressiveness[37,38] that could be directly compared to the metabolic metrics which are also continuous in nature.

**HP ¹³C-MRI detects occult prostate lesions and enables non-invasive metabolic phenotyping of multifocal disease.** HP [1-¹³C]lactate signal was observed exclusively within tumours in all patients and not in other areas of the prostate, consistent with previous reports[20,21]. Two lesions (an example is illustrated in Fig. 1) were not reported prospectively on pre-biopsy mpMRI but were detected retrospectively by correlating whole-mount histopathology with HP [1-¹³C]lactate maps. This observation corroborates previous reports[20,21] and highlights the potential added value of HP ¹³C-MRI for more accurate evaluation of the burden of multifocal disease and that could therefore be used to influence management decisions, for example using focal therapy. Furthermore, as illustrated in Fig. 1, both HP [1-¹³C]lactate signal-to-noise ratio (lactate SNR) and $k_{PL}$ were significantly elevated in lesions harbouring higher ISUP grade disease (the same was true for tumours 3 and 4 in Table 1), in agreement with a previous clinical report[21] and preclinical work in this area[39], showing the potential of HP ¹³C-MRI to enable comparative characterisation of tumour aggressiveness in multifocal PCa.

**HP [1-¹³C]lactate labelling correlates with the percentage of Gleason pattern 4 disease and tumour cellularity but not vascular permeability.** To further evaluate the relationship between

**Table 1 Histopathological and imaging characteristics of prostate tumours included in the image analysis.**

| Tumour | Final Gleason score | Final ISUP grade group | % GP4 | Mean ADC | Mean lactate SNR | Mean pyruvate SNR | Mean total carbon SNR | Mean $k_{PL}$ |
|---|---|---|---|---|---|---|---|---|
| 1 | $3 + 4 = 7$ | 2 | <5 | 1239.0 | 8 | 33 | 49 | 0.015 |
| 2 | $3 + 4 = 7$ | 2 | <5 | 1380.0 | 9 | 37 | 53 | 0.012 |
| 3 | $3 + 4 = 7$ | 2 | 10 | 921.5 | 11 | 69 | 87 | 0.009 |
| 4 | $4 + 3 = 7$ | 3 | 60 | 788.7 | 16 | 70 | 89 | 0.015 |
| 5 | $3 + 4 = 7$ | 2 | 5 | 743.6 | 12 | 21 | 39 | 0.014 |
| 6 | $3 + 3 = 6$ | 1 | 0 | 1074.0 | 6 | 19 | 27 | 0.011 |
| 7 | $3 + 4 = 7$ | 2 | 15 | 845.0 | 15 | 22 | 37 | 0.004 |
| 8 | $3 + 4 = 7$ | 2 | 30 | 721.0 | 14 | 26 | 51 | 0.005 |
| 9 | $3 + 4 = 7$ | 2 | 20 | 1060.0 | 16 | 47 | 77 | 0.006 |
| 10 | $3 + 4 = 7$ | 2 | <5 | 763.1 | 40 | 68 | 112 | 0.018 |
| 11 | $3 + 4 = 7$ | 2 | <5 | 923.3 | 10 | 15 | 29 | 0.003 |
| 12 | $4 + 3 = 7$ | 3 | 50 | 581.5 | 24 | 36 | 62 | 0.015 |
| 13 | $3 + 3 = 6$ | 1 | 0 | 762.0 | 8 | 31 | 54 | 0.009 |

Mean ADC values are presented as $10^{-6}$ mm$^2$/s, and mean kPL values are presented as s$^{-1}$.
SNR signal-to-noise ratio, ISUP International Society of Urological Pathology, ADC apparent diffusion coefficient.

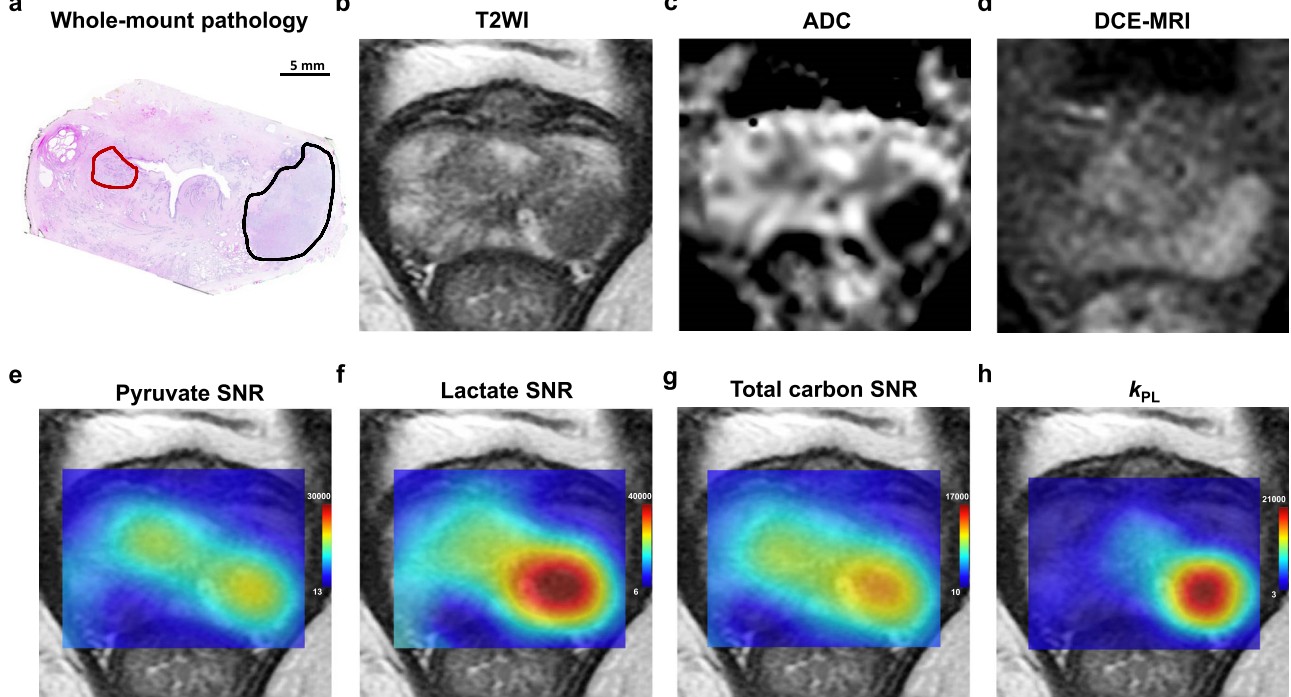

**Fig. 1 Representative example of an MR-occult transition zone tumour detected on HP $^{13}$C-MRI.** 64-year-old patient who underwent robot-assisted radical prostatectomy (patient 8 in Supplementary Table 1). **a** Post-surgical histopathological assessment confirmed the diagnosis of multifocal adenocarcinoma of the prostate. An International Society of Urological Pathology (ISUP) grade 3 target lesion was visible on $^1$H-MRI in the left peripheral zone: Prostate Imaging-Reporting and Data System (PI-RADS) 5 target lesion; black region of interest (ROI); tumour 12 in Table 1 and Supplementary Table 1). An additional $^1$H-MRI occult ISUP grade 1 lesion was present in the right transition zone: red ROI; tumour 13 in Table 1 and Supplementary Table 1. **b** Standard-of-care T$_2$-weighted MRI demonstrating a marked area of low signal intensity corresponding to the target lesion in the left peripheral zone. **c** ADC map demonstrating a corresponding focus of markedly restricted diffusion in the left peripheral zone. **d** Dynamic contrast-enhanced (DCE) MRI demonstrating the area of early enhancement in the left peripheral zone. **e** Pyruvate signal-to-noise ratio (SNR) map with two areas of high pyruvate signal, both corresponding to histopathology-confirmed tumour foci. **f** Lactate SNR map demonstrating high [1-$^{13}$C]lactate signal in the grade 3 left peripheral zone lesion. **g** Total carbon SNR map showing higher signal in the left peripheral zone tumour. **h** $k_{PL}$ map (presented as s$^{-1}$) showing a higher rate of pyruvate-to-lactate conversion in the more aggressive left peripheral zone lesion.

HP $^{13}$C-MRI metabolic parameters and tumour aggressiveness, the imaging metrics were correlated with standard-of-care pathological (%GP4) and imaging (ADC) biomarkers of clinically significant PCa. ADC maps were used to provide a clinical measure of tissue cellularity, which can differentiate clinically aggressive and indolent PCa[40]. Figure 2a and Supplementary Table 2 show the Spearman correlation analysis which revealed the presence of strong correlations between lactate SNR, %GP4 and mean ADC ($r_s = 0.65$ and $-0.69$, $P = 0.03$ and 0.02, respectively) with the latter two parameters intercorrelating negatively ($r_s = -0.62$, $P = 0.03$) as shown previously[40]. Lactate SNR also demonstrated a strong positive correlation with histopathology-derived tumour epithelial cell number ($r_s = 0.80$, $P = 0.002$; Fig. 2b, Supplementary Table 3), while no correlations

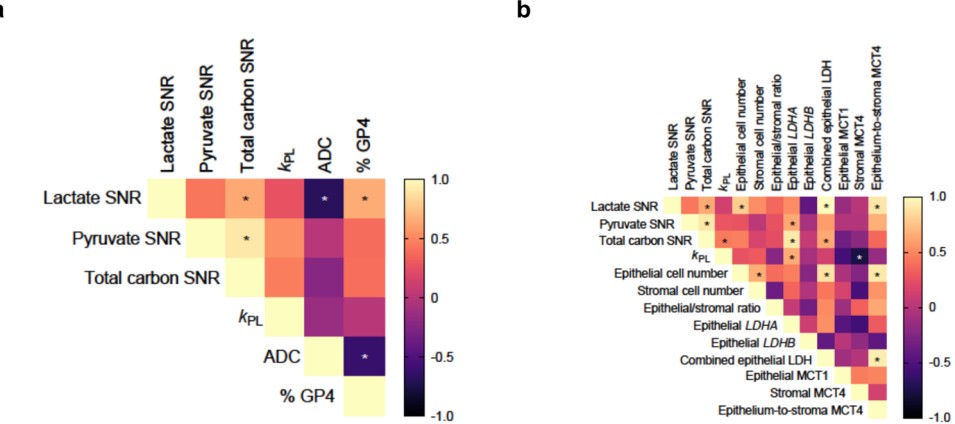

**Fig. 2 Correlation analysis of HP $^{13}$C-MRI metabolic parameters, $^{1}$H-MRI-derived apparent diffusion coefficient (ADC), %GP4, tumour cell composition, and expression of MCT1, MCT4, _LDHA_, and _LDHB_.** Heatmaps representing the correlations between HP $^{13}$C-MRI metabolic parameters and: (**a**) $^{1}$H-MRI-derived ADC and histopathology-derived %GP4; and (**b**) histopathology-derived tumour epithelial and stromal cell numbers and the ratio of tumour epithelial-to-stromal cells, and RNAscope-derived tumour epithelial _LDHA_ and _LDHB_ expression (average mRNA copies per cell), combined epithelial LDH (summed copy numbers of _LDHA_ and _LDHB_), and IHC-derived percentages of tumour and stromal cells expressing MCT1 and MCT4, respectively. For each correlation pair, the values were extracted from single ROIs encompassing the whole tumours to allow for appropriate correlation of the continuous quantitative outputs derived from different imaging and biological techniques used. Spearman's correlation analysis was used in both cases, with asterisks denoting statistically significant correlations and colour scales representing the range of individual rank correlation coefficients. Individual Spearman's rank correlation coefficients, their 95% confidence intervals, and _P_ values for each of the presented correlation pairs are listed in Supplementary Information.

were observed between lactate SNR and tumour-associated stromal cell number or tumour epithelium-to-stroma ratio derived from the same ROIs ($r_s$ = 0.52 and 0.36, $P$ = 0.07 and 0.22, respectively; Fig. 2b, Supplementary Table 3). These findings suggest that the [1-$^{13}$C]lactate signal is driven by tumour epithelial cell-specific metabolism and increases with the proportion of a more aggressive histological subtype within intermediate-risk tumours. We also observed significant positive correlations between total carbon SNR and pyruvate SNR ($r_s$ = 0.90, $P$ < 0.0001; Fig. 2a, Supplementary Table 2). Finally, we saw no significant correlations between $K^{trans}$ and HP $^{13}$C-MRI parameters (Supplementary Table 4), suggesting that the observed variation in [1-$^{13}$C]lactate labelling provided additional information to that offered by measurements of tumour perfusion and vascular permeability derived from quantitative assessment of dynamic contrast-enhanced (DCE) MRI.

**HP [1-$^{13}$C]lactate labelling correlates with the epithelium-to-stroma MCT4 ratio.** To investigate the mechanisms driving HP [1-$^{13}$C]lactate labelling in the imaged lesions, we evaluated both the overall expression and the tumour epithelial and stromal distribution of MCT1 and MCT4 on IHC (Fig. 3a, Supplementary Table 5). There was no difference in the overall expression of MCT1 and MCT4 derived from both tumour epithelial and stromal cells, although MCT1 was expressed predominantly on epithelial cells and MCT4 primarily on stromal cells (Fig. 3a, Supplementary Table 5). The epithelial MCT1 and stromal MCT4 showed a trend that was negative with all the HP $^{13}$C-MRI metabolic parameters, with a significant negative correlation observed between stromal MCT4 and $k_{PL}$ ($P$ = 0.009; Fig. 2b, Supplementary Table 3). Importantly, the tumour epithelium-to-stroma MCT4 ratio showed a positive correlation with the lactate SNR ($r_s$ = 0.90, $P$ = 0.002; Fig. 2b, Supplementary Table 3) that was stronger than the correlation between epithelial MCT4 and lactate SNR ($r_s$ = 0.36, $P$ = 0.31), demonstrating the potentially important role of metabolic compartmentalisation within the tumour in the interpretation of HP $^{13}$C-MRI.

**HP [1-$^{13}$C]lactate labelling reflects an increase in the combined _LDHA_ and _LDHB_ expression in the tumour epithelium.** To elucidate the impact of LDH expression on [1-$^{13}$C]lactate labelling, we quantified the mRNA expression of _LDHA_ and _LDHB_ in both the tumour epithelial and stromal compartments (Fig. 3e, Supplementary Table 5). There are five isoforms of LDH which contain varying numbers of LDHA and LDHB subunits: the former subunit exhibits a higher affinity towards pyruvate and therefore preferentially converts pyruvate to lactate and NADH to NAD$^+$, whereas LDHB demonstrates a higher affinity towards lactate, resulting in a preferential conversion of lactate to pyruvate, and NAD$^+$ to NADH[41]. Quantifying _LDHA_ and _LDHB_ expression separately enabled us to evaluate the relationship between isoenzyme pattern and [1-$^{13}$C]lactate labelling, and we also assessed the combined expression of the two genes since the steady-state lactate concentration is determined by the total LDH activity[42,43]. The overall _LDHA_ mRNA expression from both tumour epithelial and stromal cells was significantly lower compared to that of _LDHB_ ($P$ = 0.01), with no difference observed between epithelium- and stroma-derived _LDHA_ and _LDHB_ expression (Fig. 3e, Supplementary Table 5). While two studies by the same group[44,45] have demonstrated overexpression of LDH5 in PCa epithelial cells using IHC, the differential expression of epithelial and stromal _LDHA_ or _LDHB_ in clinical PCa samples has not been demonstrated previously, which presents an interesting area for future studies.

Tumour epithelial _LDHA_ expression correlated positively with lactate SNR, pyruvate SNR, and $k_{PL}$, with the latter two correlations reaching statistical significance ($r_s$ = 0.64 and 0.66, $P$ = 0.04 and 0.03, respectively; Fig. 2b, Supplementary Table 3). Epithelial _LDHB_ was negatively correlated with HP $^{13}$C-MRI parameters, although no correlations were significant (Supplementary Table 3). Importantly, combined LDH expression in the epithelium (_LDHA_ and _LDHB_) showed a significant correlation with lactate SNR ($r_s$ = 0.99, $P$ < 0.001; Fig. 2b, Supplementary Table 3). These results suggest that in the present cohort consisting of low- and intermediate-risk PCa, [1-$^{13}$C]lactate labelling was driven predominantly by the combined LDH expression rather than MCT1, which had been shown to play

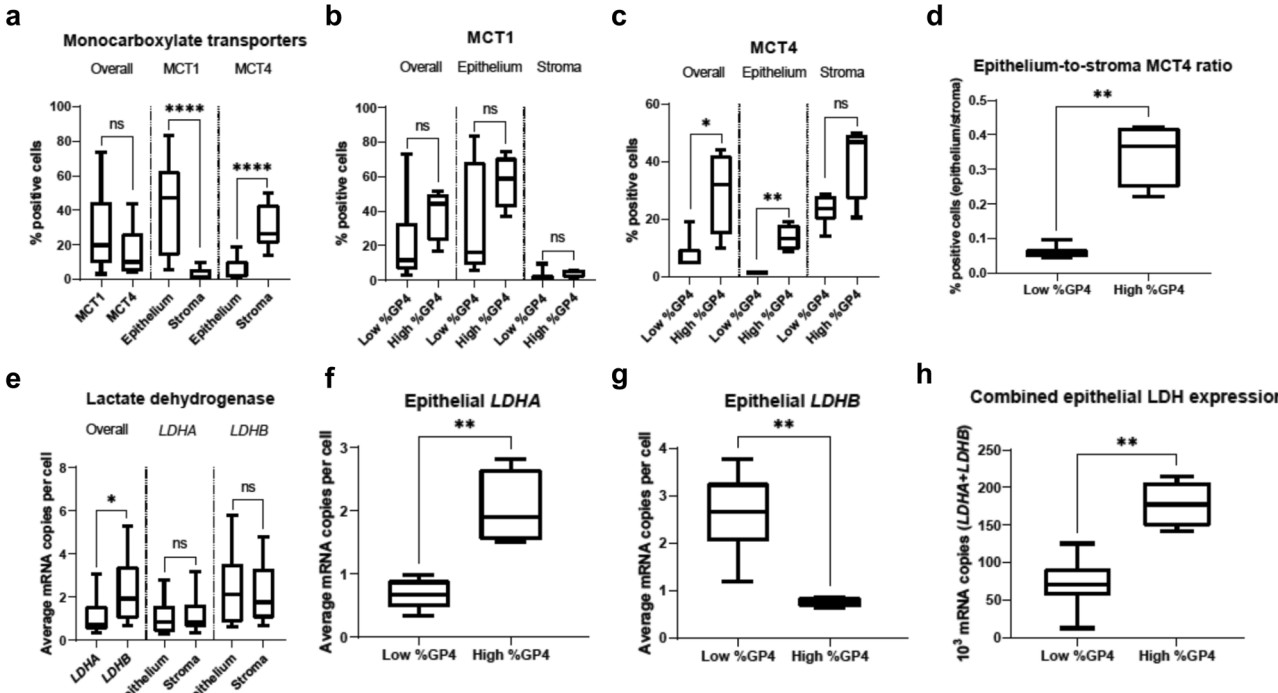

**Fig. 3 Analysis of the overall and cell-specific expression patterns of MCT1, MCT4, *LDHA*, and *LDHB* between tumours with high and low percent Gleason pattern 4. a** Box-and-whisker plots comparing the overall and cell-specific IHC-derived expression of MCT1 and MCT4, measured as percentage positive cells, in tumours imaged with HP $^{13}$C-MRI ($n = 12$ biologically independent samples). The figure also shows the overall, epithelial, and stromal expression of: (**b**) MCT1 and (**c**) MCT4, in tumours with low (≤10%) and high (>10%) %GP4. **d** Epithelium-to-stroma MCT4 ratio in tumours with low and high %GP4. Data in panels (**b–d**) were derived from $n = 10$ biologically independent samples. **e** Overall and compartmentalised RNAscope-derived expression of *LDHA* and *LDHB* in tumours imaged with HP $^{13}$C-MRI ($n = 13$ biologically independent samples). **f** *LDHA*, and (**g**) *LDHB* expression, in tumours with low and high %GP4 is also shown. **h** Combined epithelial LDH, measured as the summed *LDHA* and *LDHB*, in tumours with low and high %GP4. Data in panels (**f–h**) were derived from $n = 11$ biologically independent samples. Centre lines represent the median values, boxes denote the interquartile ranges (25th to 75th percentiles), and whiskers denote the minimum and maximum values. *$P < 0.05$; **$P < 0.01$; ***$P < 0.001$; ****$P < 0.0001$ were derived using the two-sided Mann–Whitney U test, with the exact values presented in Supplementary Table 5. Source data are provided as a Source Data file.

an important role in a previous clinical study assessing a different patient cohort with higher grade prostate tumours in comparison to the population studied here[21].

**HP [1-$^{13}$C]lactate labelling in high percentage Gleason pattern 4 is associated with increased tumour epithelial LDH expression.** To further investigate the mechanism by which the increasing %GP4 contributes to higher lactate SNR, the ISUP grade 2-3 lesions were divided into high %GP4 ($n = 4$) and low % GP4 ($n = 6$), with %GP4 of 10% chosen as a cut-off value based on previous studies[46,47]. We observed a significant increase in the overall MCT4 expression in more aggressive disease ($P = 0.02$; Fig. 3c, Supplementary Table 5), due to its increased expression on tumour epithelial cells, as demonstrated by a significant increase in the epithelium-to-stroma MCT4 ratio in high %GP4 lesions (Fig. 3d, Supplementary Table 5). The epithelium-to-stroma MCT4 ratio demonstrated very strong positive correlations with lactate SNR and combined epithelial LDH expression ($r_s = 0.90$ and 0.93, $P = 0.002$ and 0.001; Fig. 2b, Supplementary Table 3), thereby suggesting an association between [1-$^{13}$C]lactate labelling, tumour LDH activity, and compartmentalised MCT4 expression. In contrast, despite an apparent trend, no significant difference in MCT1 expression was seen between low and high %GP4 lesions (Fig. 3b, Supplementary Table 5). *LDHA* expression increased in the high %GP4 group ($P = 0.01$; Fig. 3f, Supplementary Table 5), and *LDHB* expression decreased ($P = 0.04$; Fig. 3g, Supplementary Table 5), as demonstrated by a significant increase in the *LDHA/LDHB* ratio ($P = 0.006$; Supplementary Table 5). Increased

*LDHA* and MCT4 expression in aggressive prostate lesions are in agreement with previous studies[36,48], and changes in *LDHB* have been reported previously[44,49,50]. However, the role of total LDH expression and individual isoenzyme pattern in the context of clinical hyperpolarised $^{13}$C-MRI has not been elucidated in previous work and provides an important additional measure of metabolism that can be used in future studies.

In addition, we hypothesised that the observed metabolic switch towards a more glycolytic phenotype in high %GP4 would be associated with changes in the expression of pyruvate dehydrogenase (PDH), a multi-protein complex that regulates flux through the TCA cycle by catalysing the conversion of pyruvate to acetyl coenzyme A (CoA)[51]. To test this, we imaged *PDHA1* mRNA expression in addition to *LDHA* using RNAscope (representative images shown in Supplementary Fig. 3) to generate an *LDHA/PDHA1* expression ratio as a marker of glycolytic to oxidative metabolism that had been shown previously to be elevated in metastatic prostate lesions[22]. In high %GP4 tumours, *PDHA1* was significantly lower compared to low %GP4 tumours (Supplementary Fig. 3; Supplementary Table 6), while *LDHA/PDHA1* was significantly higher (Supplementary Fig. 3; Supplementary Table 6), which provides indirect evidence for a shift from oxidative to glycolytic metabolism associated with the emergence of a more glycolytic cell population.

**Single-gland analysis of MCT and LDH expression reveals metabolic progression towards increased glycolysis in Gleason pattern 5 glands.** To assess whether the changes suggestive of

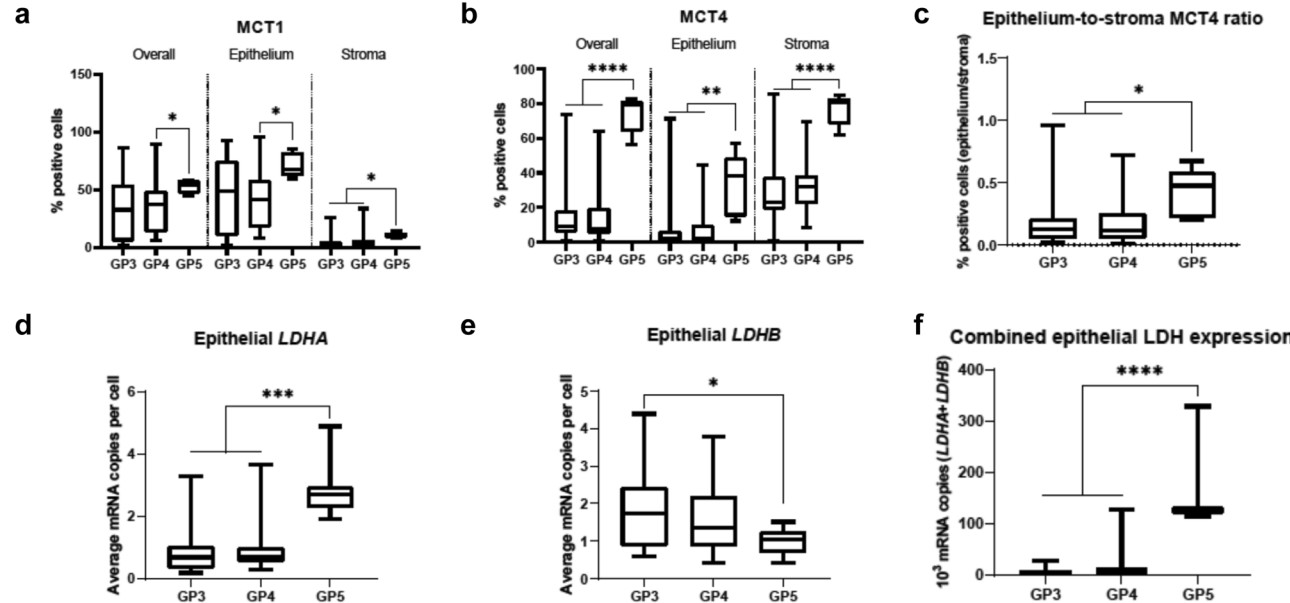

**Fig. 4 Comparison of MCT1, MCT4, *LDHA*, *LDHB*, and combined epithelial *LDHA* and *LDHB* expression in malignant glands harbouring Gleason pattern 3, 4, and 5 disease.** Box-and-whisker plots comparing the overall and cell-specific IHC-derived expression of: (**a**) MCT1, (**b**) MCT4, and (**c**) epithelium-to-stroma MCT4 ratio, derived from regions of interest (ROIs) of malignant prostate glands and adjacent stroma harbouring GP3, GP4, and GP5 disease, representing $n = 79$ biologically independent samples. The figure also shows RNAscope-derived: (**d**) *LDHA*, (**e**) *LDHB*, and (**f**) combined LDH expression across GP3, GP4, and GP5 glands, representing $n = 86$, $n = 89$, and $n = 86$ biologically independent samples per panel, respectively. Centre lines represent the median values, boxes denote the interquartile ranges (25th to 75th percentiles), and whiskers denote the minimum and maximum values. $*P < 0.05$; $**P < 0.01$, $***P < 0.001$, $****P < 0.0001$ were derived using the two-sided Mann–Whitney U test, with the exact values presented in Supplementary Table 7. Source data are provided as a Source Data file.

increased glycolytic flux in high %GP4 tumours were more pronounced in the most aggressive GP5 glands, we evaluated MCT and *LDH* expression in areas encompassing individual malignant glands and adjacent stroma comprised of pure Gleason patterns 3, 4, and 5 (Supplementary Fig. 4). There were no significant differences between any of the parameters extracted from GP3 and GP4 glands (Fig. 4; Supplementary Table 7), although the overall trend resembled that observed in low and high %GP4 disease. However, GP5 glands showed significantly higher overall MCT1 expression compared to GP4 glands due to both increased tumour epithelial and stromal expression (Fig. 4a; Supplementary Table 7), which is in agreement with previously published mRNA expression data showing increased MCT1 in high-grade disease[21]. The increase in MCT4 expression between GP5 glands and both GP3 and GP4 disease was even more prominent (Fig. 4b; Supplementary Table 7), and although there was an increase in both epithelium and stroma MCT4, this was more marked in the epithelial compartment, as shown by the epithelium-to-stroma MCT4 ratio (Fig. 4c; Supplementary Table 7). *LDHA* expression was also significantly higher in GP5 glands (Fig. 4d; Supplementary Table 7), while *LDHB* expression was significantly lower in GP5 glands compared to GP3 and GP4 glands (Fig. 4e; Supplementary Table 7). The resulting combined epithelial *LDH* expression was significantly higher in GP5 glands (Fig. 4f; Supplementary Table 7). Representative IHC and RNAscope images illustrating these findings are presented in Fig. 5.

To validate the observed increase in *LDHA* expression in GP5 tumours, we analysed the nuclear expression of HIF-1α using IHC (Supplementary Fig. 5; Supplementary Table 8). In GP5 glands, the nuclear HIF-1α expression in both tumour epithelial and stromal cells was significantly higher compared to GP3 and GP4 glands (Supplementary Fig. 5; Supplementary Table 8), which corroborates the increased expression of *LDHA* shown above, as one of the key HIF-1α target genes[52,53].

The *LDHA/PDHA1* ratio was also significantly higher in GP5 glands compared to GP3 and GP4 disease (Supplementary Fig. 3; Supplementary Table 9). Overall, these observations are consistent with the known increase in glycolysis in late-stage aggressive PCa[36,48], while also in agreement with reports in which changes in glycolysis could not differentiate GP3 and GP4 disease, suggesting a step-change nature of the emergence of a more glycolytic phenotype in GP5 disease[54–56]. These findings provide further support for the importance of increased tumour epithelial LDH expression with more aggressive disease, which has direct implications for the clinical use of HP $^{13}$C-MRI. In addition, these results demonstrate not only increased MCT expression in high-grade tumours, but also a relative redistribution of MCT4 expression from the stromal to the epithelial compartment, as seen in both Fig. 3d and Fig. 4c, which could be used as an early biomarker of an increase in tumour epithelial glycolytic flux. However, it is important to stress that while epithelial MCT4 expression increases, stromal MCT4 expression is still considerably higher in more aggressive disease.

**TCGA data from a large prostatectomy cohort corroborates the observed changes in MCT4, *LDHA*, and *LDHB* expression.** To assess the generalisability of the observed patterns in an independent prostatectomy cohort, we analysed normalised RNA-Seq by Expectation Maximisation (RSEM) data for *SLC16A1* (MCT1), *SLC16A3* (MCT4), *LDHA*, and *LDHB* obtained from prostatectomy samples of 498 patients as part of the TCGA-PRAD study[57,58]. Since one of the key results of our study was the differential [1-$^{13}$C]lactate labelling in lesions with varying %GP4, we grouped the TCGA-PRAD tumours by their primary Gleason patterns to reflect this. Consistent with our IHC findings, *SLC16A1* expression showed no difference between lesions with

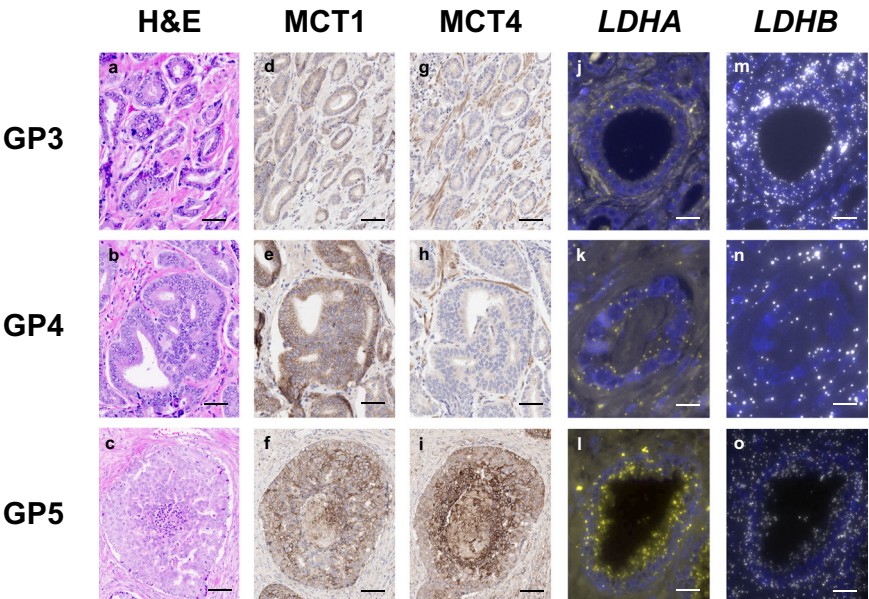

**Fig. 5 Representative H&E, immunohistochemistry, and RNAscope images demonstrating changes in the overall and cell-type-specific expression of MCT1, MCT4, *LDHA*, and *LDHB* in Gleason pattern 3, 4, and 5 disease. a–c** Digitised haematoxylin-and-eosin images (H&E) showing GP3, GP4, and GP5 adenocarcinoma of the prostate, respectively. **d–f** MCT1 immunohistochemical staining of the same glands demonstrated expression which is predominantly in tumour epithelium and is consistent across GP3, GP4, and GP5 glands. **g–i** Conversely, MCT4 immunohistochemical staining showed predominantly stromal expression in GP3 and GP4 glands, and a shift towards tumour expression in GP5 glands, an observation reflected in the increased epithelium-to-stroma MCT4 ratio detailed in the text. **j–l** RNAscope immunofluorescent staining for *LDHA* mRNA (spectrum gold) demonstrated the gradual increase in the intensity of staining in the progression to GP5 disease. **m–o** Conversely, RNAscope immunofluorescent staining for *LDHB* mRNA (spectrum white) revealed its decreasing expression in the progression to GP5 disease, while the combined LDH expression increased. Scale bars on images (**a, b, d, e, g, h**), (**c, f, i, l, o**), and (**j, k, m, n**), denote 200 μm, 100 μm, and 10 μm, respectively.

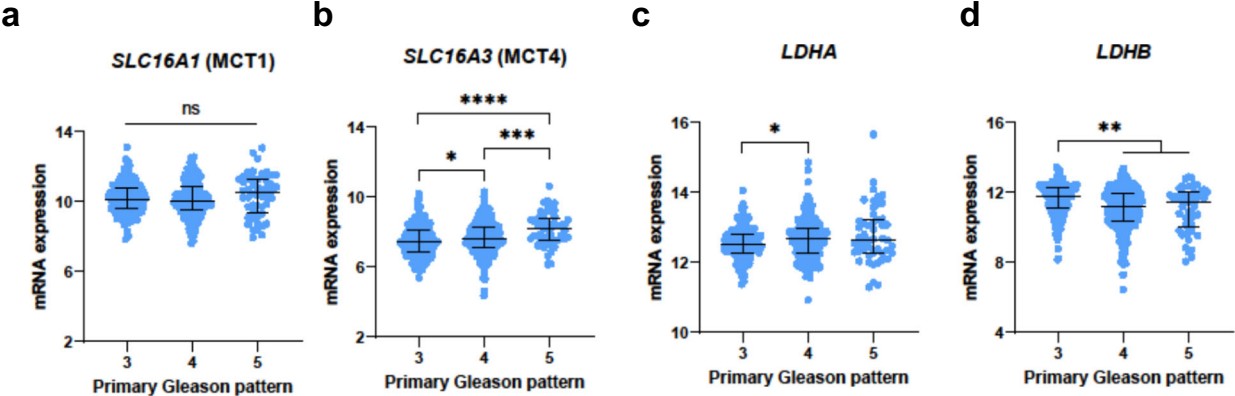

**Fig. 6 Analysis of the TCGA-PRAD RNA sequencing profiles of *SLC16A1*, *SLC16A3*, *LDHA*, and *LDHB* in prostate cancer, derived from prostatectomy samples.** Mixed box-and-whisker and scatterplots comparing the RSEM normalised for (**a**) *SLC16A1*, (**b**) *SLC16A3*, (**c**) *LDHA*, and (**d**) *LDHB* mRNA expression, in tumours with primary Gleason patterns 3, 4, and 5, representing $n = 497$ biologically independent samples. The RSEM values were [$\log_2$(value-1)] transformed. The boxes denote the median values, and the whiskers denote the interquartile ranges (25th to 75th percentile). *$P < 0.05$; **$P < 0.01$, ***$P < 0.001$, ****$P < 0.0001$ were derived using the two-sided Mann-Whitney U test, with the exact values presented in Supplementary Table 10. The raw data behind this figure were originally obtained as part of the TCGA-PRAD study and downloaded from the CBioPortal[57,58]. Source data are provided as a Source Data file.

primary GP3 and GP4 (Fig. 6a; Supplementary Table 10), while *SLC16A3* was significantly overexpressed in primary GP4 disease (Fig. 6b; Supplementary Table 10). In line with our RNAscope results, *LDHA* expression was higher and *LDHB* expression was lower in primary GP4 lesions compared to primary GP3 disease (Fig. 6c, d; Supplementary Table 10). Also, gene expression patterns in primary GP3, GP4, and GP5 lesions (as demonstrated in Fig. 6 and Supplementary Table 10), mirror those reported by our single-gland analysis presented in Fig. 4 and Supplementary

Table 7. However, statistically significant differences in mRNA expression between primary GP3-4 and GP5 lesions in the TCGA-PRAD dataset were noted for only two genes, *SLC16A3* (Fig. 6b, Supplementary Table 10) and *LDHB* (Fig. 6d, Supplementary Table 10). This may be explained by the reduced statistical power of tests that assessed the primary GP5 data due to the considerably lower number of primary GP5 tumours ($n = 50$) in the TCGA-PRAD dataset compared to primary GP3 ($n = 197$) and GP4 ($n = 250$) lesions.

## Discussion

This prospective study explored the mechanisms underpinning increased [1-$^{13}$C]lactate labelling in human PCa by correlating non-invasive HP $^{13}$C-MRI with standard-of-care imaging and pathological features of tumour aggressiveness, alongside assessment of the differential cellular expression of genes and proteins that influence lactate labelling. This included measurement of the membrane transporters of pyruvate and lactate (MCT1, MCT4) and mRNA for the enzymatic subunits that catalyse the exchange between the two molecules (*LDHA, LDHB*), within tumour and stromal cells. We also measured the expression of *PDHA1* to assess the potential for pyruvate entry into the TCA cycle and the nuclear abundance of HIF-1α, a transcription factor that drives *LDHA* expression.

Whole-mount prostatectomy specimens confirmed the ability of HP $^{13}$C-MRI to detect occult areas of PCa on standard-of-care mpMRI and showed strong correlations between [1-$^{13}$C]lactate signal, %GP4, and diffusion-weighted MRI measurements of tissue cellularity. While ADC reflects total cellularity within a region of interest, the strong relationship between lactate SNR and the histologically-derived measure of tumour epithelial cell number within a lesion demonstrates that [1-$^{13}$C]lactate labelling reports specifically on tumour epithelial cell metabolism. This finding is supported by preferential expression of MCT1 in this cellular compartment which facilitates the uptake of the hyperpolarised [1-$^{13}$C]pyruvate, as well as the increase in combined epithelial LDH and overexpression of MCT4 on tumour epithelial cells in high %GP4 disease, both of which show strong correlations with [1-$^{13}$C]lactate signal. In addition to providing a biological interpretation of HP $^{13}$C-MRI findings, these results have implications for the future clinical translation of the technique.

Previous clinical studies have demonstrated a grade-dependent increase in [1-$^{13}$C]lactate labelling in PCa[21], breast[59] and renal[60] tumours, which is also supported by the reported association between a glycolytic phenotype and poor clinical outcomes in these tumour types[36,61,62]. Here, we observed differences in [1-$^{13}$C]lactate signal in tumours of the same grade but a different prevalence of Gleason pattern 4 disease, which suggests that HP $^{13}$C-MRI can detect subtle but clinically important intra-grade metabolic heterogeneity, despite the limitations of its spatial resolution, with as little as 10% of the tumour containing GP4 disease. While the observed metabolic profile of GP3 and GP4 glands was similar, this ability to non-invasively differentiate tumours based on %GP4 will be particularly useful in the management of low-to-intermediate-risk disease on active surveillance, for which this parameter is key to determining treatment[63]. More specifically, HP $^{13}$C-MRI may improve baseline selection of patients suitable for active surveillance enrolment, where men harbouring tumours with high [1-$^{13}$C]lactate labelling could be considered for immediate radical treatment or be offered more stringent follow-up to enable timely detection of any disease progression. Furthermore, it may improve subsequent assessment on surveillance by providing a quantitative metabolic measure of tumour progression to supplement the subjective MRI-derived Prostate Cancer Radiological Estimation of Change in Sequential Evaluation (PRECISE) scoring system used in clinical practice[64]. Finally, prospective validation of our findings in larger cohorts will enable an increase in the statistical power for evaluating metabolic differences between GP3 and low %GP4 lesions, compared with high %GP4 and GP5 disease. Significant metabolic differences between these subgroups would not only further justify the suitability of this technique for active surveillance of patients with low %GP4 disease, but also underpins the clinical utility of HP $^{13}$C-MRI for non-invasive metabolic risk-stratification of PCa.

Several groups have investigated the effects of monocarboxylate transporter and lactate dehydrogenase expression and activity on [1-$^{13}$C]pyruvate metabolism[21,22,48,59,65–67]. [1-$^{13}$C]Lactate labelling has been shown to correlate significantly with MCT1 expression in breast and prostate cancer patients[21,59], and experiments in vitro have demonstrated the importance of MCT1 for [1-$^{13}$C]pyruvate-to-[1-$^{13}$C]lactate conversion in selected cell lines[67]. Increased MCT4 expression and LDH expression or activity have also been shown to be distinct features of high-grade and metastatic prostate tumours[22,36,48]. Here we have shown that the key drivers of higher [1-$^{13}$C]lactate labelling in the more aggressive subtype of intermediate-risk PCa are the combined tumour epithelial LDH (combined *LDHA* and *LDHB*), and the associated increase in tumour epithelium-to-stroma MCT4 expression, with both markers demonstrating a much stronger relationship with lactate SNR compared to epithelial MCT1 and epithelial or stromal MCT4 alone. The observed pattern indicates the emergence of a lactate-producing epithelial cell population in high %GP4 lesions, and this study has identified tissue-based biomarkers to detect this metabolic transformation such as tumour epithelial *LDHA, LDHB, PDHA1, LDHA/PDHA1* ratio, and epithelium-to-stroma MCT4 ratio. If validated in larger cohorts, these biomarkers could provide important tissue-based metabolic phenotyping assays for clinical practice in the future. HP $^{13}$C-MRI as a non-invasive tool for assessing the whole organ would be very complementary to these histopathological tools and could be used to inform on the suitability of patients with advanced disease for specific targeted treatments, including MCT1 inhibitors and glycolysis-targeting agents[68,69]. In addition, given the interrelationship between glycolysis, hypoxia, and genomic instability in high-grade PCa[34], metabolic phenotyping may be used to both identify appropriate candidates for targeted therapy[70] and evaluate treatment response[71]. More specifically, tumour [1-$^{13}$C]lactate labelling may serve as a marker of treatment response to poly (ADP-ribose) polymerase (PARP) inhibition or androgen deprivation, which both have been shown to inhibit the glycolytic activity of tumour cells[72–74].

The most aggressive GP5 glands are metabolically distinct from GP3-4 disease, with significantly higher combined epithelial LDH expression and a significant increase in both epithelial and stromal MCT1 and MCT4 expression. Like high %GP4 disease, the increase in the combined epithelial LDH expression in GP5 glands was driven primarily by *LDHA*, while *LDHB* decreased significantly, which has not been reported previously. The increase in *LDHA* expression can be explained by significantly higher nuclear HIF-1α expression in GP5 glands, consistent with the previous work[52,53,59,75]. Our results are similar to those seen in a TCGA dataset, and taken together with the previously published work in this area[22,36,74,76], these provide further support for increased glycolysis and tissue hypoxia as established features of high-grade PCa[21,34,77,78]. In summary, this work has shed light on the molecular complexity driving the glycolytic changes in prostate tumours, both between lesions of varying aggressiveness, and between tumour epithelial and stromal compartments. Expression of MCT4 and combined LDH (*LDHA* and *LDHB*) in the tumour epithelium are important in determining clinically significant intermediate-risk disease, while MCT1 expression may also play a role in more aggressive tumours.

The limitations of this study include the absence of imaging data from high-risk disease. However, a previous study by Granlund et al. has demonstrated an increased [1-$^{13}$C]lactate signal with higher grade tumours, and Chen et al. have illustrated high $k_{PL}$ in prostate cancer metastases[22]. Although the size of the cohort is relatively small, it is similar to that in other published

clinical studies involving this novel imaging technique[20–22,59,60]. Importantly, our key findings are supported by the TCGA-PRAD database. Although both mRNA expression and IHC measures were used to evaluate the key molecular drivers, RNAscope was used to quantify the spatial gene expression profile in the same serial FFPE slides used for IHC analysis, thereby enabling a more direct comparison between the two analyses in the same tumour areas.

In conclusion, this study demonstrates differential [1-13C]lactate labelling in intermediate-risk PCa with varying percentage pattern 4 components and confirms the ability of HP 13C-MRI to provide additional information to grade-dependent disease differentiation used as part of standard of care. We provide evidence that increasing [1-13C]lactate labelling in more aggressive disease is a reflection of tumour epithelial cell metabolism, as opposed to stromal metabolism, as demonstrated by an increase in the combined tumour epithelial LDH expression and epithelium-to-stroma MCT4 ratio. We have also demonstrated a possible reciprocal relationship between tumour glycolysis and oxidative metabolism, as evidenced by the *LDHA/PDHA1* expression ratio, and high nuclear HIF-1α expression in malignant glands of the most aggressive histological pattern, which may explain the increase in *LDHA* expression. These findings have demonstrated the ability of HP 13C-MRI to non-invasively differentiate indolent from aggressive prostate tumours based on their characteristic metabolic features. If validated in larger patient cohorts, these findings have translational potential to address important unmet clinical questions for patients with PCa using a novel imaging modality.

## Methods

**Patient recruitment and ethics**. This prospective study was approved by the institutional review board (National Research Ethics Service Committee East of England, Cambridge South, Research Ethics Committee number 16/EE/0205) under the MISSION-Prostate protocol (Molecular Imaging and Spectroscopy with Stable Isotopes in Oncology and Neurology—Imaging metabolism in prostate). The study included consecutive male patients (median age 65 years, interquartile range 62–68 years) with MR-visible (>1 cm) histologically proven PCa (index lesion ISUP grade group ≥2) scheduled for radical prostatectomy. All patients provided written consent to participate in this study and were recruited between May 2018 and February 2020 without any participant compensation. The exclusion criteria were as follows: any prior treatment for PCa, presence of pelvic metalwork, acute major illness, and any clinical contraindication to MRI including renal impairment as defined by GFR < 30 ml/min.

The flowchart of the recruitment process is presented in Supplementary Fig. 1. In two patients, we were unable to achieve adequate HP 13C-MRI quality, evidenced by a tumour-derived total carbon SNR of <5.0 (Supplementary Fig. 2), leading us to exclude their images from the HP 13C-MRI analysis. One patient, who was recruited into this study in February 2020 and underwent a successful HP 13C-MRI study, had his surgery postponed until June 2020 due to the COVID-19 pandemic restrictions and was prescribed interval androgen deprivation therapy (ADT). Due to the pronounced impact of ADT on multiple metabolic pathways[79], prostatectomy tissue samples obtained from this patient were excluded from the IHC and RNAscope analysis, and one lesion was excluded from the correlation analysis due to difficult %GP4 assessment. The final cohort characteristics are summarised in Supplementary Table 1.

**Hyperpolarised [1-13C]pyruvate MRI acquisition**. Samples containing 1.47 g of [1-13C]pyruvic acid (Sigma Aldrich) and 15 mM electron paramagnetic agent (EPA) were hyperpolarised using a clinical hyperpolarizer (SPINlab; 5 T Research Circle Technology, GE Healthcare, Waukesha WI, USA) by microwave irradiation at 139 GHz at ∼0.8 K for ∼3 h followed by rapid dissolution in 38 mL of super-heated sterile water and filtration to remove EPA to a concentration below ≤3 μM[59]. HP 13C-MR images were acquired on a clinical 3 T MR system (MR750, GE Healthcare, Waukesha WI, USA) using a bespoke 1H/13C endorectal receive coil[80]. Radiofrequency pulses with a nominal flip angle of 15° were applied to acquire a 20 × 20 cm2 FOV with a matrix size of 32 × 32 and a temporal resolution of 4 s for 20 time points. Images had a true in-plane resolution of 12.5 × 12.5 mm2 and were reconstructed with a resolution of 128 × 128. The scans were acquired using either a dynamic coronal Iterative Decomposition with Echo Asymmetry and Least-squares estimation (IDEAL) spiral chemical shift imaging (CSI) sequence[81] with a repetition time (TR) of 0.5 s or an 8-step IDEAL cycle or using a spectral-spatial (SpSp) pulse with TR = 2 s and flip angles of 15° on pyruvate and 40° on lactate[82]

**HP 13C-MRI data analysis**. The IDEAL and SpSp imaging data were reconstructed in MATLAB (MathWorks, Natick, MA, USA)[59]. For the analysis of metabolite ratios integrated over time, the complex data were summed over all time points prior to coil combination to minimise noise propagation. Regions of interest (ROIs) were drawn manually on co-registered T2-weighted images using whole-mount histopathology maps as a reference using OsiriX 10.0 (Pixmeo SARL, Bernex, Switzerland). The signal-to-noise ratio (SNR$_{metabolite}$) was calculated from maps of pyruvate and lactate within each ROI as follows:

$$\text{SNR}_{\text{metabolite}} = \frac{\text{mean}(\text{SI}_{\text{ROI}}) - \text{mean}(\text{SI}_{\text{noise}})}{\sqrt{2}\text{S.D.}(\text{SI}_{\text{noise}})} \quad (1)$$

where mean (SI$_{ROI}$) is the mean signal intensity in the ROI, and the mean S.D. of SI$_{noise}$ are computed over the ROI containing background only. The factor of $\sqrt{2}$ accounts for the narrowed Rayleigh distribution of magnitude noise, with an approximate adjustment for the use of multiple receivers. Total carbon SNR (pyruvate + lactate + other 13C-labelled metabolite peaks) maps were generated using Eq. (1), the summed carbon signal at each point, and the noise distribution from the pyruvate image at the final timepoint of each series, when the signal had invariably disappeared. In addition, we calculated the apparent reaction rate constant for the exchange of the HP 13C label between pyruvate and lactate ($k_{PL}$) using a two-site exchange model using a frequency-domain approach and linear least-squares fitting.

**Proton MRI acquisition and analysis**. Standard-of-care MRI of the prostate was performed on the same scanner either prior to patient recruitment into this study or immediately after HP 13C MR images were acquired. The protocol included axial T1-weighted fast spin echo (FSE), high-resolution T2-weighted 2D fast recovery fast spin echo (FRFSE, echo time [TE] 98-107 ms, field-of-view [FOV] 18 × 18 cm, acquisition matrix 320–384 × 256, slice thickness 3 mm with 0 mm gap, 3 signal averages, repetition time [TR] 3000–5000 ms, echo train length 16, receiver bandwidth ± 31.25 or ± 41.67 kHz), diffusion-weighted imaging (DWI, spin-echo echo-planar imaging pulse sequence with b values of 150, 550, 750, 1000, and 1400, with a separate high b value acquisition of 2000 s/mm2), and dynamic contrast-enhanced MRI (DCE-MRI, axial 3D fast spoiled gradient echo [FSPGR], TR/TE 4.1/1.8 ms, FOV 24 × 24 cm2, following bolus injection of gadobutrol [Gadovist, Bayer Healthcare, Berlin, Germany] via a power injector, rate 3 mL/s [dose 0.1 mmol/kg], temporal resolution 7 s)[83]. Apparent diffusion coefficient (ADC) maps were calculated automatically, with the mean ADC values extracted from ROIs drawn manually by a fellowship-trained uroradiologist with 13 years of experience in reporting prostate MRI (TB) using T2-weighted as a reference. The same ROIs were used for extracting $K^{trans}$ values, with DCE-MRI analysis performed using in-house-developed MATLAB software that was used to generate B1 maps[59]. These maps were then transferred to MIStar (Apollo Medical Imaging, Melbourne, Australia) to generate B1-corrected T1 maps, to perform motion correction of the DCE-MRI data using a 3D affine model, and for pharmacokinetic modelling using the standard Tofts model[59].

**Histopathological assessment**. Whole-mount surgical sections were stained with haematoxylin and eosin (H&E)[84], with an experienced genitourinary pathologist (AYW) outlining the foci of PCa, assigning Gleason grades, and evaluating the percent Gleason pattern 4 (%GP4) within grade group 2 and 3 lesions. Tumour outlines on whole-mount H&E maps were used as a reference when drawing ROIs on HP 13C-MRI as described above. The same ROIs were transposed onto IHC and RNAscope images for the analyses described below in the corresponding sections. In addition, the same pathologist used whole-mount H&E maps to draw standardised random ROIs within larger PCa foci that encompassed malignant glands and adjacent stroma consisting of clear Gleason pattern 3, 4, and 5 disease (Supplementary Fig. 4). These ROIs were also transposed onto IHC and RNAscope images for the below analyses.

**Immunohistochemistry**. In all patients, IHC staining for MCT1, MCT4, and HIF-1α was performed on formalin-fixed, paraffin-embedded (FFPE) prostatectomy tumour blocks using Leica's Polymer Refine Detection System (DS9800) in combination with their Bond automated system (Leica Biosystems Newcastle Ltd, Newcastle, UK). Sections were cut to 4 μm thickness and baked for 1 h at 60 °C ahead of deparaffinisation and rehydration, as standard, on the ST5020 Multistainer (Leica Biosystems). Subsequent immunohistochemical staining was carried out on Leica's automated Bond III platform (Leica Biosystems) in conjunction with their Polymer Refine Detection System (Cat. No. DS9800, Leica Biosystems). Sections stained for HIF-1 alpha (Cat. No. ab51608, Abcam, Cambridge, UK) were pre-treated with Epitope Retrieval Solution 1 (Cat. No. AR9961, Leica Biosystems) and those stained for MCT1 (Cat. No. HPA003324, Atlas Antibodies, Bromma, Sweden) and MCT4 (Cat. No. HPA021451, Atlas Antibodies, Bromma, Sweden) were pre-treated with Epitope Retrieval Solution 2 (Cat. No. AR9640, Leica Biosystems). Incubation was for 20 min at 99 °C. Antibodies were diluted to 23.36 μg/ml, 0.2 μg/ml and 0.6 μg/ml respectively. Endogenous peroxidase activity was quenched using 3–4% (v/v) hydrogen peroxide and primary antibody was detected using Anti-rabbit Poly-HRP-IgG (<25 μg/mL; part of Leica Biosystems Polymer Refine Detection System) containing 10% (v/v) animal serum in tris-buffered

**Table 2 Channels, fluorophores, and catalogue numbers of *LDHA*, *LDHB*, and *PDHA1* RNAscope probes used in this study.**

| Target name (Channel) | Detecting Fluorophore | Catalogue Number |
|---|---|---|
| Hs-LDHA-C1 | Opal 570 | 487818 |
| Hs-LDHB-C2 | Opal 650 | 531278-C2 |
| Hs-PDHA1-C3 | Opal 620 | 892218-C3 |

saline/0.09% ProClin 950.The complex was visualised using 66 mM 3,3′-Diaminobenzidine tetrahydrochloride hydrate in a stabiliser solution and ≤0.1% (v/v) Hydrogen Peroxide. DAB Enhancer (Cat. No. AR9432, Leica Biosystems) was used to intensify the signal. Cell nuclei were counterstained with <0.1% haematoxylin.

HALO v3.2.1851.266 (Indica Labs, Albuquerque, NM, USA) Membrane v1.7 module for MCT1 and multiplex IHC v2.3.4 module for MCT4 and HIF-1α (clone EP1215Y) were used for automated analysis of scanned sections. Optical densities for weak, moderate and strong stains used for the automated quantitative analysis of scanned sections were: MCT1, 0.1602, 0.2302, 0.4037; MCT4, 0.1967, 0.2544, 0.3872; HIF-1α nuclear 0.1958, 0.7522, 0.885 and cytoplasm 0.1856, 0.4064, 0.5973. The number of cells showing weak, moderate, and strong staining for MCT1 and MCT4 were summed and divided by the total number of cells within an ROI to obtain the percentage of positive cells for MCT1 and MCT4 expression. The nuclear HIF-1α expression was prioritised over its overall and cytoplasmic expression due to its physiological activity as a transcription factor and quantified by dividing the number of cells showing weak, moderate, and strong nuclear staining for HIF-1α by the total number of cells within an ROI. Due to the post-hoc nature of the analysis aimed at validating the observed *LDHA* mRNA expression patterns, not all cases had sufficient amount of tissue remaining for HIF-1α staining.

A random forest classifier was used to distinguish between the two tissue classes: epithelial and stromal cells, with an example image presented in Supplementary Fig. 6. Randomly picked annotations of the different tissue classes from a selection of images were used to train the classifier, with the full analysis pipeline described in a separate methodological study[85], where a similar approach was applied to RNAscope images. The outputs of this classifier enabled us to quantify the overall, epithelium- and stroma-derived expression of the described proteins, as well as the number of epithelial and stromal cells within an ROI, which we used for the correlation analysis presented in Fig. 2 and Supplementary Table 3.

**RNAscope**. RNAscope is an in situ hybridisation technology that allows for subcellular detection, visualisation, and quantification of target mRNA in intact formalin-fixed paraffin-embedded tissue samples[86]. The double-Z design of target probes utilises a pool of oligonucleotide probes and amplifies target-specific signals but not the background noise from non-specific hybridisation[86]. RNAscope has been validated against conventional single-sell RNAseq[87] and bulk qPCR[88], which require tissue destruction and therefore do not allow the spatial assessment of mRNA expression or its co-localisation with other modalities such as immunohistochemistry or imaging, which was a critical element of our study. Here, sections were cut to 4 μm thickness and baked for 1 h at 60ºC before loading onto a Bond RX instrument (Leica Biosystems Newcastle Ltd, Newcastle, UK). Slides were deparaffinized and rehydrated on board prior to pre-treatments with Epitope Retrieval Solution 2 (Cat No. AR9640, Leica Biosystems) at 95 °C for 15 min, and ACD Enzyme from the Multiplex Reagent kit at 40 °C for 15 min. Probes (listed in Table 2) were visualised using Opal fluorophores diluted to 1:1000 using RNAscope® LS Multiplex TSA Buffer. Probe hybridisation, signal amplification, and detection were all performed on the Bond Rx according to the ACD protocol. Slides were removed from the Bond Rx and mounted using Prolong Diamond (Cat. No. P36965, ThermoFisher Scientific, Watham, MA, USA).

In all patients, simultaneous detection of human *LDHA*, *LDHB*, and *PDHA1* was performed on FFPE sections using Advanced Cell Diagnostics (ACD, Bio-Techne, Abingdon, UK) RNAscope 2.5 LS Multiplex Reagent Kit (Cat No. 322800), and RNAscope 2.5 LS probes (ACD, Hayward, CA, USA). The slides were imaged on the AxioScan (Carl-Zeiss-Stiftung, Stuttgart, Germany) to create whole-slide images. Images were captured at ×40 magnification, with a resolution of 0.25 microns per pixel. HALO v3.2.1851.266 and the FISH v2.2.0 module were used for the automated analysis of scanned RNAscope images. A similar fluorescent random forest tissue classifier[85] was used to differentiate the overall, epithelial, and stromal mRNA expression within the same ROIs used for IHC analysis. mRNA expression of *LDHA, LDHB, and PDHA1* was quantified as the average mRNA copy number per cell. The combined epithelial LDH expression was quantified as a sum of the total number of *LDHA* and *LDHB* copies detected in epithelial cells within the defined tumour ROIs.

**TCGA-PRAD data analysis**. To validate the prospective findings obtained from this study, we analysed RSEM data from the TCGA-PRAD study of prostatectomy samples (accessed through CBioPortal and the Broad Institute Firehose

Legacy)[57,58]. The genes included in the analysis were *SLC16A1, SLC16A3, LDHA,* and *LDHB*. The normality of the data was assessed using the D'Agostino-Pearson test, with the following intergroup comparison of individual gene expression in tumours with primary Gleason patterns 3, 4, and 5, performed using the Mann-Whitney U test.

**Statistics and reproducibility**. Statistical analyses were conducted using Graph-Pad Prism (version 9.0.2, GraphPad Software, San Diego, CA, USA). Normal distribution of the data was assessed using the D'Agostino-Pearson test (threshold $P \geq 0.05$). Correlation analysis was conducted using the Spearman's rank correlation test since at least one variable was always non-normal. Intergroup differences as part of the analysis of IHC, RNAscope, and TCGA-PRAD RNAseq data were measured using the Mann-Whitney U test. All statistical tests were two-tailed, and $P$ values below 0.05 were considered significant. No multiplicity correction was applied, and therefore all significant tests should be interpreted as exploratory rather than confirmatory. All experiments were independent and standalone.

**Reporting summary**. Further information on research design is available in the Nature Research Reporting Summary linked to this article.

## Data availability
The authors declare that the data supporting the findings of this study are available within the article and its Supplementary Information. The open-source TCGA-PRAD data used in this study are available on cBioPortal [https://www.cbioportal.org/study/summary?id=prad_tcga]. Source data are provided with this paper.

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

## Acknowledgements

This study was supported by Prostate Cancer UK (PCUK; Grant PA14-012) and Cancer Research UK (CRUK; Grants C19212/A27150, C19212/A16628, C197/A29580, C197/A17242). N.S. is supported by the Gates Cambridge Trust. Additional support provided from the Cancer Research UK Cambridge Centre, the National Institute of Health Research (NIHR) Cambridge Biomedical Research Centre, the Cambridge Experimental Cancer Medicine Centre, a Wellcome Trust Strategic Award, Addenbrooke's Charitable Trust, and Cambridge University Hospitals National Health Service Foundation Trust.

## Author contributions

N.S., M.A.M., K.M.B., T.B. and F.A.G. formulated the research idea, designed and planned the study. N.S., M.A.M., I.G.M., K.M.B., T.B. and F.A.G. wrote the article. M.A.M., A.Y.W., A.J.V.B., C.B., A.B.G., J.J., J.L.M., J.D.K., A.N.P., F.J.L.R., R.F.S. and M.J.G. performed imaging and tissue-based studies and analysed the data. N.S., A.F., M.J.L., B.W.L., V.J.G., T.B. and FAG coordinated patient recruitment and oversaw their clinical management. T.B. and F.A.G. contributed equally to this manuscript.

## Competing interests

The authors declare the following competing interests: F.A.G. has research support from GE Healthcare, grants from GlaxoSmithKline, and has consulted for AstraZeneca on behalf of the University of Cambridge. K.M.B. has research support from GE Healthcare. F.J.L.R. and R.S.S. are GE Healthcare employees. The remaining authors declare no competing interests.
