## [Peer Review File · Nature Communications]

REVIEWER COMMENTS

Reviewer #1 (Remarks to the Author): Expert in prostate cancer MRI

This is a well written manuscript, with a sound methodologic approach, elucidating the mechanisms underpinning the ability of hyperpolarized (HP) ¹³C MRI to differentiate aggressive from indolent prostate cancer based on increased HP [1-¹³C]lactate labelling after injection of HP [1-¹³C]pyruvate. Specifically, it was demonstrated that in a low/intermediate-risk population that HP [1-¹³C]lactate labeling correlated with the % of Gleason pattern 4 in the tumor. From the standpoint of interpreting the HP [1-¹³C]pyruvate imaging results, another important finding was that HP-¹³C-MRI predominantly measures metabolism in the epithelial rather than stromal compartment of the tumor. This finding took advantage of a new spatial transcriptomics technique, RNAscope. The authors also demonstrate that in tumors with greater than 10% Gleason pattern 4 had a significant increase in HP [1-¹³C]lactate labelling versus those with < 10% pattern 4, and mechanistically this increased [1-¹³C]lactate labeling was associated with significantly increased epithelial mRNA expression of the enzyme lactate dehydrogenase (LDHA and LDHB combined) with LDHA increasing and LDHB decreasing, and increased epithelial MCT4 and epithelial to stromal MCT4 ratio in high %Gleason 4 tumors. It was also interesting that the switch to a more glycolytic phenotype in high % Gleason 4 tumors was associated with a reduction in PDHA (oxidative metabolism). These findings provide further evidence that HP [1-¹³C]pyruvate MRI will be useful in the discriminating aggressive from indolent disease in a low to intermediate risk population and thereby improve the selection of appropriate candidates for active surveillance or more aggressive treatment. Finally, the data supporting a further increase in glycolysis and the significant increase in HIF-1 α and MCT1 in Gleason 5 glands provides support for the role of HP ¹³C-MRI in later stage disease, and for selecting targeted therapy. The findings of this study both complement the existing literature and provide strong new evidence that Hyperpolarized [1-¹³C]pyruvate predominately visualizes increased glycolytic metabolism in the tumor epithelium with increasing pathological grade. Therefore, this study would be of significance to the fields of prostate cancer and hyperpolarized ¹³C MR imaging and could improve patient treatment.

As pointed out by the authors, the limitations of this study are the absence of imaging data from high-risk prostate cancer and the small patient cohort. These concerns are somewhat mitigated by the fact that small patient cohorts are endemic for studies involving this new metabolic imaging technique, and that there are two small studies showing increased [1-¹³C]lactate in high grade localized and metastatic prostate cancer. The authors also attempted to generalize their findings using data from the Cancer Genome Atlas. The statement that “We validated these findings in a large TCGA cohort showing significant changes in MCT4, LDHA, and LDHB between tumours of varying Gleason score” is overstated and the manuscript would be improved by modifying this statement. There were no significant changes in MCT1 and only a significant difference between MCT4 and LDHA between Gleason 6 or 7 versus Gleason 9 disease in the TCGA data analyses shown in Fig 6. Since one of the main findings of this study, was that HP [1-¹³C]lactate labeling correlated with the % of Gleason pattern 4 in the tumor, it might have been better to analyze the TCGA data based on cancers having primary 3, 4 and 5 grade diseases rather than by Gleason score.

Reviewer #2 (Remarks to the Author): Expert in 13C MRI

Prostate serum antigen (PSA) is the standard biomarker used to assess PCa, but its non-specificity for clinically significant PCa has led to over utilization of prostate biopsy and diagnosis of indolent disease. Indeed, the number needed to treat (NNT) with PSA testing to avert one death has been cited to be as high as 48 cases. Though 1H MRI avoids unnecessary biopsies and reduces over detection of indolent diseases, as the introduction states, it has limitations in clinically significant PCa. The goal in this study is to utilize 13C MRI using hyperpolarized pyruvic acid and monitoring the conversion kinetics to lactate in low-, intermediate and high grade PCa.

Key findings are: 1) lactate labeling is correlated to MCT and LDH expression in the epithelium; 2) Validation of these findings with TCGA database. This study shows the potential of 13C MRI with hyperpolarized probes to differentiate between indolent and aggressive PCa based on labeling kinetics. Though limited to 10 patients, given the challenges in technology and current situation of the pandemic, the data is valuable and can be accepted for publications. Minor suggestions are given below for consideration.

1) Consider this statement instead of lines 52-53. "In addition to unnecessary diagnoses, however, our current screening modalities (PSA and systematic prostate biopsy) also can miss clinically significant prostate cancer".

2) I would consider eliminating the phrase talking about biochemical recurrence. You are focusing on the ability of 13C-pyruvate to risk stratify clinically prostate cancer before intervention, not to predict recurrence after intervention. (lines 58-59)

3) I would suggest mentioning the similar metabolic profile between GP3 and GP4 and rationalize it as demonstrating metabolic heterogeneity in GP4 based on percentage. Although the study may not be powered enough to detect this subgroup difference, I would imagine that the metabolic profile (MCT4 epithelium to stroma ratio and LDHA epithelial expression) would be similar between GP5 and high percentage GP4 lesions. On the other hand, GP3 may behave similar to low percentage GP4. If validated in larger studies, this not only provides justification for active surveillance for low volume GP4 but also demonstrates the utility of this imaging study in identifying appropriate candidates for active surveillance, as you state below.(lines 440-441).

4) Would mention that validation of findings are needed with larger sample sizes using this imaging modality. (after lines 527-529)

Reviewer #3 (Remarks to the Author):Expert in 13C MRI

This is an interesting report on additional clinical studies, applying HP 13C-MRI technology to the field of oncology, particular prostate cancer. Although the number of patients studied is modest, new information on biomolecular analysis via in situ hybridization of HP-imaged prostate tumors is provided and correlated with HP 13C imaging results.

Overall, the work appears to be well executed, innovative, and worthy of publication.

However, some specific questions, issues and inconsistencies exist that should be addressed and clarified.

Line 55 – the “and” should probably be “or” or “and/or”

Lines 95-98: At least one of the cited references does not appear to support the statement made here, having a contrary conclusion (i.e., Pertega-Gomes 2011,. Another reference does not even involve the evaluation of CAF's (Pertega-Gomes 2015) and probably should therefore be excluded from the references cited. Recommend that authors closely review the literature they cite to support the contention made in this particular sentence.

Line 121 – the parameter of “number of tumour epithelial cells” is listed in the Introduction and elsewhere as being significantly correlated with HP13C-lactate, but how exactly tumor epithelial cell number is determined and defined in specimens does not seem to be provided in any detail.

Line 183 – “=0,80” should be “=0.80”

Figure 2: To this reviewer, the 3 graphics in this figure are hard to read. Suggest simplifying information within two graphics. Additionally, why is “Total Carbon SNR” only included in 1 of the 3 graphics? This parameter should be included across the 3 graphics (or 2, if simplified).

Minor – The capital “A”, “B”, etc labelings within the graphics should also be used in the figure legend (not the small-case “a”, “b”, etc provided).

Lines 231-260: The method of LDHA and LDHB mRNA assessment that underlies the statements in this section should be discussed (either in the main text or in Supplemental material). For instance, how does RNAscope analysis compare to laser-capture microdissection mRNA analysis and IHC analysis.

Relatedly, some discussion seems warranted about the authors' analysis showing that LDHA expression is not significantly greater in tumor epithelial compartment compared to tumor stromal compartment, and how this compares/contrasts with previous reports in the literature.

Figure 3 – As in Figure 2, the “A”, “B”, “C” labeling on the graphics should also be used in the figure legend.

Lines 295-297: The statement here that “changes in LDHB have not been previously reported in this context” does not seem to be entirely accurate to this reviewer. Consider the findings reported in Leiblich et al. Oncogene 2006 and Giatromanolaki et al. Cancer Biol Ther 2012.

Lines 299-307: How was LDHA measured for this ratio of LDHA/PDHA1? By RNAscope, as was PDHA1? -- or some other method. This info should be included within the text of the manuscript here in this section.

Also, in the corresponding Suppl Figure 2, the histology in the A and B panels should be of “Low %GP4” and “High %GP4” specimens, to better match the bar-graph in C. Also, the D and E graphs in Suppl Fig2 seem to be presented slightly backwards, should not GP3 vs GP4 vs PG5 data be presented first. Then present the more focused-in (i.e., more subtle distinction) difference between Low %GP4 vs High %GP4 data.

Line 353-356: Would be helpful to reference Figure 3D panel somewhere in the text of this sentence. Authors should also probably acknowledge that, although the relative % of MCT4 goes up in High %GP4 tumour epithelium, its expression is still much higher in tumour stroma overall.

Figure 4: 1) Same issue with the “A”, “B”, “C” labeling not being consistent with legend labeling.

2) In panel F, why are the units on the Y-axis different than those in panel D and E? These axes labels should be the same if possible.

Line 482-485: This sentence and its associated list of references should probably include refer #72 (Aggarwal et al. Eur Urol 2017), published case report of HP 13C-lactate decrease with androgen deprivation therapy in a prostate cancer patient.

Line 499 – The phrase “across tumour cell subtypes” is confusing. If the authors are referring to stromal cells vs epithelial cells within a tumour, they should probably rephrase this to make it clear.

Figures 5 & 6: Same issue with the “A”, “B”, “C” labeling not being consistent with legend labeling “a”, “b”, “c”, etc.

Reviewer #4 (Remarks to the Author): Expert in prostate cancer and gene expression analyses

Prostate cancer development and progression are accompanied by dynamic metabolic changes, with early-stage tumors exhibiting primarily TCA cycle/mitochondrial respiration but high-grade/aggressive tumors manifesting increased aerobic glycolysis and glycolytic flux. In this manuscript, authors employed hyperpolarized ¹³C-MRI (HP-¹³C-MRI) to monitor metabolic changes in patient prostate tumors. They correlated ¹³C-lactate labeling signal with patients' Gleason pattern 4 (%GP4) in attempt to distinguish indolent vs. progressive diseases in a low/intermediate risk population of prostate cancer patients (n=10 patients). The major finding is that HP-¹³C-MRI mainly measures metabolism in the tumor epithelium (rather than stroma), and the HP-¹³C-MRI signal correlated with the LDHA/LDHB expression levels and with the ratio of epithelium-to-stroma MCT4 (a lactate transporter). Authors conclude that HP-¹³C-MRI can metabolically phenotype clinically progressive prostate tumors. Although the study has potential clinical significance in metabolically phenotyping indolent vs. aggressive prostate tumors, the conceptual advance appears to be limited, and relatively small numbers of lesions/patients and some technical issues diminish the overall significance.

1. Previous studies in the field have utilized ¹³C-pyruvate as the substrate to gauge the rate of glycolytic (pyruvate-to-lactate) flux (e.g., STM, 5:198ra108; 2013; CCR 24:3137-48, 2018; Cell Metab. 31:105-114, 2020). As such studies have reported the potential utility of HP-¹³C-MRI in non-invasively assessing the aggressiveness of prostate tumors, the conceptual novelty of the current study seems to be incremental. Also, the concept that advanced, high-grade and aggressive prostate tumors assume a more glycolytic metabolism is now well-established.

2. Authors claimed some of the above-mentioned earlier studies as 'anecdotal'; however, the overall numbers of patients (n=10) and tumors (n=13) in this study were small. In fact, only 11 of the 13 tumors had matched ¹³C, IHC and RNAScope data. And only 9 of the tumors were of ISUP Grade 2. Importantly, all data presented are correlative without any functional validation.

3. There are confusions and problems about presented data. For example, Fig. 1 was based on data from a 64-year-old patient who underwent RARP. Suppl. Table 2 indicated two 64 year-old patients: pt. 2 (who had tumors 3 and 4) and pt. 10 (who had tumor 14 and 15). However, Fig. 1 legend stated that the two tumor lesions in this patient were tumor 12 and 13. Another example were the RNAScope images in Suppl. Fig. 2a-b: the stated GP3 appeared to be more like a GP4 lesion (this reviewer was trained as a pathologist).

4. Also, while data in Fig. 1 is clear in demonstrating the utility of lactate SNR/pyruvate SNR (kPL) in pinpointing the more advanced tumor lesion, it's unclear how many other high-grade foci exhibited similar glycolytic changes. From the tables (Table 1; Suppl. Table 2), it appears that there were only 2 tumors that had the ISUP grade 2.

5. Were the Correlation plots in Fig. 2 derived from just one tumor or from data pooled from several tumors/patients? The small numbers of samples may not robustly support the conclusions associated with Fig. 2.

6. Some quantitative data in Fig. 3B-C showed apparent trend of differences, which were nevertheless statistically insignificant, likely due to, again, small numbers of tumors available for analysis.

7. The nuclear staining of HIF1 α was not clear at all (Suppl. Fig. 4) and the statement of 'high nuclear HIF-1 α expression in high-grade prostate lesions (Discussion) is unconvincing and not supported by the data.

8. Both Introduction and Discussion are unnecessarily long.

Response to reviewers' comments (NCOMMS-21-25779)

Reviewer #1 - Expert in prostate cancer MRI

1. This is a well written manuscript, with a sound methodologic approach, elucidating the mechanisms underpinning the ability of hyperpolarized (HP) ¹³C MRI to differentiate aggressive from indolent prostate cancer based on increased HP [1-¹³C]lactate labelling after injection of HP [1-¹³C]pyruvate. Specifically, it was demonstrated that in a low/intermediate-risk population that HP [1-¹³C]lactate labeling correlated with the % of Gleason pattern 4 in the tumor. From the standpoint of interpreting the HP [1-¹³C]pyruvate imaging results, another important finding was that HP-¹³C-MRI predominantly measures metabolism in the epithelial rather than stromal compartment of the tumor. This finding took advantage of a new spatial transcriptomics technique, RNAscope. The authors also demonstrate that in tumors with greater than 10% Gleason pattern 4 had a significant increase in HP [1-¹³C]lactate labelling versus those with < 10% pattern 4, and mechanistically this increased [1-¹³C]lactate labeling was associated with significantly increased epithelial mRNA expression of the enzyme lactate dehydrogenase (LDHA and LDHB combined) with LDHA increasing and LDHB decreasing, and increased epithelial MCT4 and epithelial to stromal MCT4 ratio in high %Gleason 4 tumors. It was also interesting that the switch to a more glycolytic phenotype in high % Gleason 4 tumors was associated with a reduction in PDHA (oxidative metabolism). These findings provide further evidence that HP [1-¹³C]pyruvate MRI will be useful in the discriminating aggressive from indolent disease in a low to intermediate risk population and thereby improve the selection of appropriate candidates for active surveillance or more aggressive treatment. Finally, the data supporting a further increase in glycolysis and the significant increase in HIF-1 α and MCT1 in Gleason 5 glands provides support for the role of HP ¹³C-MRI in later stage disease, and for selecting targeted therapy. The findings of this study both complement the existing literature and provide strong new evidence that Hyperpolarized [1-¹³C]pyruvate predominately visualizes increased glycolytic metabolism in the tumor epithelium with increasing pathological grade. Therefore, this study would be of significance to the fields of prostate cancer and hyperpolarized ¹³C MR imaging and could improve patient treatment.

We thank the Reviewer for their positive feedback and insight into the potential significance of this study.

2. As pointed out by the authors, the limitations of this study are the absence of imaging data from high-risk prostate cancer and the small patient cohort. These concerns are somewhat mitigated by the fact that small patient cohorts are endemic for studies involving this new metabolic imaging technique, and that there are two small studies showing increased [1-¹³C]lactate in high grade localized and metastatic prostate cancer.

We thank the Reviewer for highlighting the above limitation of the study, which we acknowledge in the relevant section of the Discussion. Reviewer 2 has also rightly pointed out the technological challenges of the new technique, which were exacerbated by the inability to prospectively recruit patients during the early stages of the ongoing pandemic. In line with comments 3 and 4 from Reviewer 2, we have provided specific hypotheses that warrant further testing as part of future studies enrolling larger patient cohorts.

3. The authors also attempted to generalize their findings using data from the Cancer Genome Atlas. The statement that “We validated these findings in a large TCGA cohort showing significant changes in MCT4, LDHA, and LDHB between tumours of varying Gleason score” is overstated and the manuscript would be improved by modifying this statement. There were no significant changes in MCT1 and only a significant difference between MCT4 and LDHA between Gleason 6 or 7 versus Gleason 9 disease in the TCGA data analyses shown in Fig 6. Since one of the main findings of this study, was that HP [1-¹³C]lactate labeling correlated with the % of Gleason pattern 4 in the tumor, it might have been better to analyze the TCGA data based on cancers having primary 3, 4 and 5 grade diseases rather than by Gleason score.

We thank the Reviewer for this very sensible comment. We have removed the reference to validation and agree that the attempt to generalise our findings would be more appropriate when conducting the TCGA analysis by grouping lesions based on primary Gleason patterns, rather than the summed Gleason scores. The results of the revised analysis now read as follows:

Since one of the key results of our study was the differential [1-¹³C]lactate labelling in lesions with varying %GP4, we grouped the TCGA-PRAD tumours by their primary Gleason patterns to reflect this. Consistent with our IHC findings, *SLC16A1* expression showed no difference between lesions with primary GP3 and GP4 (Figure 6A; Supplementary Table 11), while *SLC16A3* was significantly overexpressed in primary GP4 disease (Figure 6B; Supplementary Table 11). In line with our RNAscope results, *LDHA* expression was higher and *LDHB* expression was lower in primary GP4 lesions compared to primary GP3 disease (Figure 6C-D; Supplementary Table 11). Also, gene expression patterns in primary GP3, GP4, and GP5 lesions (as demonstrated in Figure 6 and Supplementary Table 11), mirror those reported by our single-gland analysis presented in Figure 4 and Supplementary Table 8. However, statistically significant differences in mRNA expression between primary GP3-4 and GP5 lesions in the TCGA-PRAD dataset were noted for only two genes: *SLC16A3* (Figure 6B, Supplementary Table 11) and *LDHB* (Figure 6D, Supplementary Table 11). This may be explained by the reduced statistical power of tests that assessed the primary GP5 data, due to the lower number of primary GP5 tumours (n = 50) in the TCGA-PRAD dataset compared to primary GP3 (n = 197) and GP4 (n = 250) lesions.

Reviewer #2 – Expert in ¹³C MRI

Prostate serum antigen (PSA) is the standard biomarker used to assess PCa, but its non-specificity for clinically significant PCa has led to over utilization of prostate biopsy and diagnosis of indolent disease. Indeed, the number needed to treat (NNT) with PSA testing to avert one death has been cited to be as high as 48 cases. Though 1H MRI avoids unnecessary biopsies and reduces over detection of indolent diseases, as the introduction states, it has limitations in clinically significant PCa. The goal in this study is to utilize 13C MRI using hyperpolarized pyruvic acid and monitoring the conversion kinetics to lactate in low-, intermediate and high grade PCa.

Key findings are: 1) lactate labeling is correlated to MCT and LDH expression in the epithelium; 2) Validation of these findings with TCGA database. This study shows the potential of 13C MRI with hyperpolarized probes to differentiate between indolent and aggressive PCa based on labeling kinetics. Though limited to 10 patients, given the challenges in technology and current situation of the pandemic, the data is valuable and can be accepted for publications.

We thank the Reviewer for their positive feedback and insightful comments that follow.

Minor suggestions are given below for consideration.

1) Consider this statement instead of lines 52-53. "In addition to unnecessary diagnoses, however, our current screening modalities (PSA and systematic prostate biopsy) also can miss clinically significant prostate cancer".

We have now revised this and agree that the addition of this statement would improve the introductory section of this manuscript.

2) I would consider eliminating the phrase talking about biochemical recurrence. You are focusing on the ability of 13C-pyruvate to risk stratify clinically prostate cancer before intervention, not to predict recurrence after intervention. (lines 58-59)

We have now removed this phrase from the manuscript.

3) I would suggest mentioning the similar metabolic profile between GP3 and GP4 and rationalize it as demonstrating metabolic heterogeneity in GP4 based on percentage. Although the study may not be powered enough to detect this subgroup difference, I would imagine that the metabolic profile (MCT4 epithelium to stroma ratio and LDHA epithelial expression) would be similar between GP5 and high percentage GP4 lesions. On the other hand, GP3 may behave similar to low percentage GP4. If validated in larger studies, this not only provides justification for active surveillance for low volume GP4 but also demonstrates the utility of this imaging study in identifying appropriate candidates for active surveillance, as you state below.(lines 440-441).

We thank the Reviewer for this insightful comment. We agree that adding this important point to the Discussion section would help navigate future studies using HP ¹³C-MRI in larger cohorts, by providing a biologically and clinically relevant hypothesis that supports the key findings presented in this manuscript. The metabolic variation in lactate formation from GP3 to GP5 is likely to represent a continuum and therefore the metabolic phenotype should provide complementary information to conventional histopathological analysis and could be used to better stratify patients in the future. The following has been added to the text:

Finally, prospective validation of our findings in larger cohorts will enable an increase in the statistical power for evaluating metabolic differences between the following subgroups: GP3 and low %GP4 lesions, compared to high %GP4 and GP5 disease. Significant metabolic differences between these subgroups would not only justify the suitability of this technique for active surveillance of patients with low %GP4 disease, but also underpin the clinical utility of HP ¹³C-MRI for non-invasive metabolic risk-stratification of PCa.

4) Would mention that validation of findings are needed with larger sample sizes using this imaging modality. (after lines 527-529)

We have now highlighted the need for validating the observed findings in larger patient cohorts in the revised manuscript. This will be important for translating HP ¹³C-MRI as a routine clinical test. The following has been added to the text:

If validated in larger patients cohorts, these findings have translational potential to address important unmet clinical questions for patients with PCa using a novel imaging modality.

Reviewer #3 – Expert in ¹³C MRI

This is an interesting report on additional clinical studies, applying HP 13C-MRI technology to the field of oncology, particular prostate cancer. Although the number of patients studied is modest, new information on biomolecular analysis via in situ hybridization of HP-imaged prostate tumors is provided and correlated with HP 13C imaging results.

Overall, the work appears to be well executed, innovative, and worthy of publication. However, some specific questions, issues and inconsistencies exist that should be addressed and clarified.

We thank the Reviewer for their positive feedback and very helpful comments that follow.

1. Line 55 – the “and” should probably be “or” or “and/or”

This has now been amended.

2. Lines 95-98: At least one of the cited references does not appear to support the statement made here, having a contrary conclusion (i.e., Pertega-Gomes 2011,. Another reference does not even involve the evaluation of CAF's (Pertega-Gomes 2015) and probably should therefore be excluded from the references cited. Recommend that authors closely review the literature they cite to support the contention made in this particular sentence.

We thank the Reviewer for bringing this to our attention. We acknowledge that the above references do not support this specific statement; the 2011 reference reflected earlier work of the same group, and the 2015 study did not discern between epithelial and stromal expression of monocarboxylate transporters. We have now substituted these references with the following more appropriate clinical studies (one by the same group from 2014) that are supportive citations for the comments made:

25. **Fiaschi, T. *et al.* Reciprocal Metabolic Reprogramming through Lactate Shuttle Coordinately Influences Tumor-Stroma Interplay. *Cancer Res.* 72, 5130–5140 (2012).**
26. **Sanità, P. *et al.* Tumor-stroma metabolic relationship based on lactate shuttle can sustain prostate cancer progression. *BMC Cancer* 14, (2014).**
27. **Pérttega-Gomes, N. *et al.* A lactate shuttle system between tumour and stromal cells is associated with poor prognosis in prostate cancer. *BMC Cancer* 14, 352 (2014).**
28. **Andersen, S. *et al.* Organized metabolic crime in prostate cancer: The coexpression of MCT1 in tumor and MCT4 in stroma is an independent**

prognosticator for biochemical failure. *Urol. Oncol. Semin. Orig. Investig.* (2015).

3. Line 121 – the parameter of “number of tumour epithelial cells” is listed in the Introduction and elsewhere as being significantly correlated with HP13C-lactate, but how exactly tumor epithelial cell number is determined and defined in specimens does not seem to be provided in any detail.

We thank the Reviewer for highlighting this and agree that this important methodological element of this study requires further clarification. We have added an overview of the random forest classifier used for these purposes to the Methods section of the manuscript, supplementing it with a new Supplementary Figure 5. The full pipeline of training the classifier and applying it to RNAscope images was previously described by the member of our team in reference 85, which we have now added to the text to facilitate the reproducibility of our findings. The amended text now reads as follows:

A random forest classifier was used to distinguish between the two tissue classes: epithelial and stromal cells, with an example image presented in Supplementary Figure 5. Randomly picked annotations of the different tissue classes from a selection of images were used to train the classifier, with the full analysis pipeline detailed previously⁸⁵, where a similar approach was applied to RNAscope images. The outputs of this classifier enabled us to quantify the overall, epithelium-, and stroma-derived expression of the described proteins, as well as derive the number of epithelial and stromal cells within an ROI, which we used for the correlation analysis presented in Figure 2 and Supplementary Table 4.

4. Line 183 – “=0,80” should be “=0.80”

This has now been amended.

5. Figure 2: To this reviewer, the 3 graphics in this figure are hard to read. Suggest simplifying information within two graphics. Additionally, why is “Total Carbon SNR” only included in 1 of the 3 graphics? This parameter should be included across the 3 graphics (or 2, if simplified).

Minor – The capital “A”, “B”, etc labelings within the graphics should also be used in the figure legend (not the small-case “a”, “b”, etc provided).

We thank the Reviewer for their useful observation. We have now combined Figure 2B-C in a single Figure 2B, supported by the new Supplementary Table 4. In the original version of the figure, we did not include total carbon SNR in two out of the three graphics because of its strong significant correlation with pyruvate SNR and k_{PL} – keeping it would make these graphics even less reader-friendly by

including a parameter that would add no additional information to the other two metrics. However, we have now explained this omission in the revised text. We have also corrected the labelling as requested.

6. Lines 231-260: The method of LDHA and LDHB mRNA assessment that underlies the statements in this section should be discussed (either in the main text or in Supplemental material). For instance, how does RNAscope analysis compare to laser-capture microdissection mRNA analysis and IHC analysis.

We thank the Reviewer for raising this interesting point. As also highlighted by Reviewer #1, RNAscope is a new spatial transcriptomics method that deserves a separate discussion to ensure that the readers can compare it with other techniques, such as those mentioned by the Reviewer. RNAscope probes all the tissue slide to be analysed simultaneously with limited manipulation of the tissue. Microdissection is very operator dependent both in terms of defining the tissue to be dissected, as well as the skills required to undertake this technically demanding procedure. However, RNAscope is limited to using only four probes simultaneously and is therefore very targeted, whereas dissected tissue samples can undergo full RNAseq analysis. We have now included a paragraph in the Supplementary Information to address this as listed below, providing some key references that can be accessed by the readers for further information:

RNAscope is a novel *in situ* hybridization technology that allows for subcellular detection, visualisation, and quantification of target mRNA in intact formalin-fixed paraffin-embedded tissue samples¹. The double-Z design of target probes utilises a pool of oligonucleotide probes and amplifies target-specific signals but not the background noise from non-specific hybridization¹. RNAscope has been validated against conventional single-cell RNAseq² and bulk qPCR³, which require tissue destruction and therefore do not allow the spatial assessment of mRNA expression or its co-localisation with other modalities such as immunohistochemistry, which was a critical element of our study.

7. Relatedly, some discussion seems warranted about the authors' analysis showing that LDHA expression is not significantly greater in tumor epithelial compartment compared to tumor stromal compartment, and how this compares/contrasts with previous reports in the literature.

We thank the Reviewer for raising this interesting question. We are unaware of any previous studies that have reported epithelial and stromal LDHA and LDHB mRNA expression in prostate cancer. However, the Sivridis group (one study mentioned by the Reviewer in their comment 9) have used IHC to show high epithelial LDHA and stromal LDHB expression in clinical prostate samples, with comparisons being made at the level of summed Gleason scores rather than primary Gleason patterns.

We have now added this to the revised paper and believe that this presents an interesting area for future studies, particularly given that mRNA levels and protein content often show discrepant results. The revised text reads as follows:

While two studies by the same group^{44,45} have demonstrated overexpression of LDH5 in PCa epithelial cells using IHC, the differential expression of epithelial and stromal *LDHA* or *LDHB* in clinical PCa samples has not been demonstrated previously, which presents an interesting area for future studies.

8. Figure 3 – As in Figure 2, the “A”, “B”, “C” labeling on the graphics should also be used in the figure legend.

We have amended this across all figures.

9. Lines 295-297: The statement here that “changes in LDHB have not been previously reported in this context” does not seem to be entirely accurate to this reviewer. Consider the findings reported in Leiblich et al. *Oncogene* 2006 and Giatromanolaki et al. *Cancer Biol Ther* 2012.

We thank the Reviewer for providing these useful references, which we have used to amend this statement. We have also stressed that the role of total LDH expression and isoenzyme pattern has not been studied previously in the context of clinical hyperpolarised ¹³C-MRI, which was the original intent of this statement. When reviewing the cited literature in line with this Reviewer’s comment 2, we also identified that Bok *et al.* (*Cancers* 2019) had shown the reduction of *LDHB* mRNA expression in murine models of aggressive prostate cancer. We have cited this reference in other parts of this manuscript and have now acknowledged it in this section as well. From a biological perspective, the production of lactate is determined by the total LDH at chemical equilibrium, and therefore this metric of total LDH provides an important additional measure of metabolism that can be used in future studies. The revised text reads as follows:

Increased *LDHA* and MCT4 expression in aggressive prostate lesions is in agreement with previous studies^{36,48}, and changes in *LDHB* have been reported previously^{44,49,50}. However, the role of total LDH expression and individual isoenzyme pattern in the context of clinical hyperpolarised ¹³C-MRI has not been elucidated in previous studies and provides an important additional measure of metabolism that can be used in future studies.

10. Lines 299-307: How was LDHA measured for this ratio of LDHA/PDHA1? By RNAscope, as was PDHA1? -- or some other method. This info should be included within the text of the manuscript here in this section.

Also, in the corresponding Suppl Figure 2, the histology in the A and B panels should be of “Low %GP4” and “High %GP4” specimens, to better match the bar-graph in C. Also,

the D and E graphs in Suppl Fig2 seem to be presented slightly backwards, should not GP3 vs GP4 vs PG5 data be presented first. Then present the more focused-in (i.e., more subtle distinction) difference between Low %GP4 vs High %GP4 data.

In this study, all mRNA measurements were performed using RNAscope (*LDHA*, *LDHB*, and *PDHA1*) and therefore the ratio of *LDHA/PDHA1* is based upon the same approach acquired from the same tissue specimens. We have now clarified this in this section as requested.

Regarding Supplementary Figure 2, we have clarified that panels A and B contain representative images of *PDHA1* mRNA expression in GP3 and GP4 glands obtained from tumours with low %GP4 and high %GP4. To further improve the readability of this figure, we have added “low %GP4” and “high% GP4” labels on panels A and B, respectively. Specific tumour identifiers that these panels refer to are now also specified in the figure legend. In terms of panels D and E, while we fully agree with the Reviewer that it would make sense to present the subtle distinction later, we chose this particular sequence to mirror our manuscript, where low% GP4 compared to high %GP4 data are discussed first since they are underpinned by the clinical hyperpolarised imaging. Hence, the reader may first want to appreciate low %GP4 vs. high %GP4 data (panels A-D) and only then move to GP3-5 analysis (panel E), which in our view is consistent with the overall flow of this manuscript. If the Reviewer feels strongly that this should be reversed, we will revise the figure.

11. Line 353-356: Would be helpful to reference Figure 3D panel somewhere in the text of this sentence. Authors should also probably acknowledge that, although the relative % of MCT4 goes up in High %GP4 tumour epithelium, its expression is still much higher in tumour stroma overall.

We have now referenced both Figure 3D (whole-tumour analysis) and Figure 4C (single-gland analysis) in this sentence, while also adding a separate sentence acknowledging the observation of the relative difference in MCT4 expression between epithelium and stroma, which we agree is important to highlight. The revised text reads as follows:

In addition, these results demonstrate not only increased MCT expression in high-grade tumours, but also a relative redistribution of MCT4 expression from the stromal to the epithelial compartment seen in both Figure 3D and Figure 4C, which could be used as an early biomarker of an increase in tumour epithelial glycolytic flux. However, it is important to stress that while epithelial MCT4 expression increases, stromal MCT4 expression is still considerably higher in more aggressive disease.

12. Figure 4: 1) Same issue with the "A", "B", "C" labeling not being consistent with legend labeling.
2) In panel F, why are the units on the Y-axis different than those in panel D and E? These axes labels should be the same if possible.

These have now been amended.

13. Line 482-485: This sentence and its associated list of references should probably include refer #72 (Aggarwal et al. Eur Urol 2017), published case report of HP 13C-lactate decrease with androgen deprivation therapy in a prostate cancer patient.

We thank the Reviewer for spotting this and have now supported this sentence with this reference, which is directly relevant to the point discussed.

14. Line 499 – The phrase "across tumour cell subtypes" is confusing. If the authors are referring to stromal cells vs epithelial cells within a tumour, they should probably rephrase this to make it clear.

We have rephrased this sentence for better clarity.

15. Figures 5 & 6: Same issue with the "A", "B", "C" labeling not being consistent with legend labeling "a", "b", "c", etc.

This has now been amended across all figures.

Reviewer #4 - Expert in prostate cancer and gene expression analyses

Prostate cancer development and progression are accompanied by dynamic metabolic changes, with early-stage tumors exhibiting primarily TCA cycle/mitochondrial respiration but high-grade/aggressive tumors manifesting increased aerobic glycolysis and glycolytic flux. In this manuscript, authors employed hyperpolarized ^{13}C -MRI (HP- ^{13}C -MRI) to monitor metabolic changes in patient prostate tumors. They correlated ^{13}C -lactate labeling signal with patients' Gleason pattern 4 (%GP4) in attempt to distinguish indolent vs. progressive diseases in a low/intermediate risk population of prostate cancer patients (n=10 patients). The major finding is that HP- ^{13}C -MRI mainly measures metabolism in the tumor epithelium (rather than stroma), and the HP- ^{13}C -MRI signal correlated with the LDHA/LDHB expression levels and with the ratio of epithelium-to-stroma MCT4 (a lactate transporter). Authors conclude that HP- ^{13}C -MRI can metabolically phenotype clinically progressive prostate tumors. Although the study has potential clinical significance in metabolically phenotyping indolent vs. aggressive prostate tumors, the conceptual advance appears to be limited, and relatively small numbers of lesions/patients and some technical issues diminish the overall significance.

We thank the Reviewer for their feedback and useful observations that follow.

1. Previous studies in the field have utilized ^{13}C -pyruvate as the substrate to gauge the rate of glycolytic (pyruvate-to-lactate) flux (e.g., STM, 5:198ra108; 2013; CCR 24:3137-48, 2018; Cell Metab. 31:105-114, 2020). As such studies have reported the potential utility of HP- ^{13}C -MRI in non-invasively assessing the aggressiveness of prostate tumors, the conceptual novelty of the current study seems to be incremental. Also, the concept that advanced, high-grade and aggressive prostate tumors assume a more glycolytic metabolism is now well-established.

We agree with the Reviewer that the increased glycolytic profile of high-grade prostate tumours is indeed well-established. However, both the biological mechanisms driving $[1-^{13}\text{C}]$ lactate labelling in human prostate cancer and the specific clinical applications of HP ^{13}C -MRI are much less established. While building on the two important clinical reports referenced by the Reviewer (STM, 5:198ra108; 2013 and Cell Metab. 31:105-114, 2020), this study introduces several novel concepts that have not been reported in the context of clinical HP ^{13}C -MRI, including: 1) the ability of the technique to offer intra-grade metabolic phenotyping of lesions with different %GP4, which has a clear translational benefit for the rapidly evolving field of prostate cancer active surveillance; 2) the specific cellular origin of $[1-^{13}\text{C}]$ lactate labelling and the resulting significance of metabolic compartmentalization for imaging pyruvate metabolism; 3) the connection between $[1-^{13}\text{C}]$ lactate labelling and the emergence of a glycolytic epithelial cell subpopulation within lesions of high %GP4, as evidenced by the differential expression of monocarboxylate transporters and total lactate dehydrogenase. We

believe that these important findings provide further insights into the previously unestablished aspects of clinical HP ¹³C-MRI in prostate cancer and provide important data for the future direction of its clinical translation.

2. Authors claimed some of the above-mentioned earlier studies as 'anecdotal'; however, the overall numbers of patients (n=10) and tumors (n=13) in this study were small. In fact, only 11 of the 13 tumors had matched ¹³C, IHC and RNAScope data. And only 9 of the tumors were of ISUP Grade 2. Importantly, all data presented are correlative without any functional validation.

We thank the Reviewer for highlighting this and agree that the use of the word "anecdotal" in this sentence may indeed be misinterpreted as diminishing the significance of the previous studies, which is not our intention. The word "reports" in this sentence refers to individual cases within these publications (rather than the studies as a whole) where biopsy-proven tumour foci were detected on HP ¹³C-MRI but not on standard-of-care mpMRI (see the previous sentence). These cases (one in each study) can indeed be considered anecdotal, just as those identified by our own work. To avoid any misinterpretation, we have removed the word "anecdotal" from this sentence.

3. There are confusions and problems about presented data. For example, Fig. 1 was based on data from a 64-year-old patient who underwent RARP. Suppl. Table 2 indicated two 64 year-old patients: pt. 2 (who had tumors 3 and 4) and pt. 10 (who had tumor 14 and 15). However, Fig. 1 legend stated that the two tumor lesions in this patient were tumor 12 and 13. Another example were the RNAScope images in Suppl. Fig. 2a-b: the stated GP3 appeared to be more like a GP4 lesion (this reviewer was trained as a pathologist).

We thank the Reviewer for raising these questions. Fig. 1 is indeed based on the findings obtained from Patient 10 in Supplementary Table 2, with the corresponding tumour numbers being 14 and 15. Supplementary Table 2 lists patients and their corresponding lesions (n = 15) in the order they were recruited into this study and scanned using HP ¹³C-MRI. Conversely, Table 1 only lists lesions that were included in the image analysis (n = 13); these lesions were renumbered consecutively after tumours 7 and 8 from Supplementary Table 2 were excluded from the image analysis. Therefore, tumours 14 and 15 from patient 10 in Supplementary Table 2 became tumours 12 and 13 in Table 1 which is confusing. We have now amended Supplementary Table 2 as follows: patients 4 and 5 (tumours 4 and 5) from the previous version are now patients 9 and 10 (tumours 14 and 15) in the new version. Hence, all other tumour numbers are now exactly matched between Table 1 and Supplementary Table 2. Finally, to further improve the readability of Fig. 1 legend, we have added specific references to Supplementary Table 2 for both the patient information and tumour numbers.

With regards to Supplementary Figure 2a-b, we have now clarified in the figure legend that these images are not suitable for diagnostic purposes given their high magnification and fluorescent nature. The purpose of this figure is to demonstrate the differential patterns of *PDHA1* mRNA expression against DAPI at the level of individual cells, and the typical cytoarchitecture used for clinical grading is not easily discernable at this level. However, these images were obtained from regions-of-interest representing Gleason pattern 3 and 4 disease respectively, which were outlined by the expert genitourinary pathologist who is an author on this paper and who also was a member of the ISUP 2019 Consensus Conference Working Group, co-developing the most recent ISUP guidelines on prostate cancer grading.

4. Also, while data in Fig. 1 is clear in demonstrating the utility of lactate SNR/pyruvate SNR (k_{PL}) in pinpointing the more advanced tumor lesion, it's unclear how many other high-grade foci exhibited similar glycolytic changes. From the tables (Table 1; Suppl. Table 2), it appears that there were only 2 tumors that had the ISUP grade 2.

As described in the manuscript, due to the high prevalence of ISUP grade 2 disease, which is typical for a surgical cohort, this study focused on assessing the utility of HP ^{13}C -MRI for intragrade rather than intergrade metabolic phenotyping, which is a method that has been demonstrated in a prior study mentioned by the Reviewer (Cell Metab. 31:105-114, 2020). However, tumours 3 and 4 from Table 1 (both imaged in patient 2 from Supplementary Table 2) demonstrate similar intergrade metabolic differences as those presented in Fig. 1. We have now added a reference to this case in the manuscript as follows:

Furthermore, as illustrated in Figure 1, both HP [$1\text{-}^{13}\text{C}$]lactate signal-to-noise ratio (lactate SNR) and k_{PL} were significantly elevated in lesions harbouring higher ISUP grade disease (the same was true for tumours 3 and 4 in Table 1), in agreement with a previous clinical report²¹ and preclinical work in this area³⁹, showing the potential of HP ^{13}C -MRI to enable comparative characterisation of tumour aggressiveness in multifocal PCa.

Finally, as shown in Table 1 and Supplementary Table 2, of 13 lesions included in the image analysis, ISUP grade 2 was assigned to 9 tumours.

5. Were the Correlation plots in Fig. 2 derived from just one tumor or from data pooled from several tumors/patients? The small numbers of samples may not robustly support the conclusions associated with Fig. 2.

For each correlation pair underlying Fig. 2, the values were derived from single regions-of-interest encompassing the whole tumours to allow for appropriate correlation of the continuous quantitative outputs derived from the different imaging and biological techniques used. This statement has now been added to the Fig. 2 legend. While the number of samples in this study is indeed relatively small,

it is comparable to that of the two previous clinical reports in prostate cancer (STM, 5:198ra108; 2013 and Cell Metab. 31:105-114, 2020); see Reviewer 1 comment 2 above. The second of these papers (Cell Metab. 31:105-114, 2020) used an equivalent correlation analysis to support some of the conclusions made by the authors. We feel our approach aligns with previous work in the field and may help identify additional, previously unexplored associations that may further our understanding of the molecular drivers and clinical surrogates of HP ¹³C-MRI-derived metabolic values.

6. Some quantitative data in Fig. 3B-C showed apparent trend of differences, which were nevertheless statistically insignificant, likely due to, again, small numbers of tumors available for analysis.

We agree with the Reviewer that these panels indeed demonstrate an apparent trend of differences that did not reach the level of statistical significance. We have now emphasized this in the manuscript, while avoiding overstating the importance of this observation, particularly given that this figure shows other statistically significant findings with the same statistical power. The amended text now reads as follows:

In contrast, despite an apparent trend, no significant difference in MCT1 expression was seen between low and high %GP4 lesions (Figure 3B, Supplementary Table 6).

7. The nuclear staining of HIF1 α was not clear at all (Suppl. Fig. 4) and the statement of 'high nuclear HIF-1 α expression in high-grade prostate lesions (Discussion) is unconvincing and not supported by the data.

We agree with the Reviewer that HIF-1 α nuclear staining patterns could be presented more clearly in this figure. We have now amended the constituent panels by: 1) magnifying the original images in panels A and B to ensure a better visual representation of individual glands; 2) replacing panel C with an image of another Gleason pattern 5 gland from the same lesion that does not have a prominent necrotic component that may confuse the readers; 3) magnifying black-box panels to the level of individual nuclei to improve the visualisation of quantitative results presented in panel D. We have also amended the statement in the Discussion to make it more consistent with the nature of our data. This now reads as follows:

We have also demonstrated a possible reciprocal relationship between tumour glycolysis and oxidative metabolism, as evidenced by the *LDHA/PDHA1* expression ratio, and high nuclear HIF-1 α expression in malignant glands of the most aggressive histological pattern, which may explain the increase in *LDHA* expression.

8. Both Introduction and Discussion are unnecessarily long.

We have included background information on prostate cancer, imaging, and this new technique to provide sufficient context that will make it easier to highlight the relevance of this study to the diverse readership of this multidisciplinary Journal. Reviewers 1-3 have asked us to add several important concepts to both sections and we feel that reducing the text of these sections may make it more difficult to read. Finally, in its current form, the manuscript is within the word limits specified by the Journal, but we are happy to take Editorial advice on this.

REVIEWERS' COMMENTS

Reviewer #1 (Remarks to the Author):

The authors have appropriately addressed all the prior critiques, and due to the significance of the findings of this manuscript to the fields of prostate cancer and hyperpolarized ^{13}C MR imaging should be published in Nature Communications.

Reviewer #4 (Remarks to the Author):

Reviewers have satisfactorily addressed my concerns.

Response to reviewers' comments (NCOMMS-21-25779A)

Reviewer #1 (Remarks to the Author):

The authors have appropriately addressed all the prior critiques, and due to the significance of the findings of this manuscript to the fields of prostate cancer and hyperpolarized ¹³C MR imaging should be published in Nature Communications.

We thank the reviewer for their kind feedback.

Reviewer #4 (Remarks to the Author):

Reviewers have satisfactorily addressed my concerns.

We thank the reviewer for their comments that helped us improve this manuscript.